# Theory predicts 2D chiral polaritons based on achiral Fabry–Pérot cavities using apparent circular dichroism

Andrew H. Salij [1], Randall H. Goldsmith[2] & Roel Tempelaar [1]✉

Realizing polariton states with high levels of chirality offers exciting prospects for quantum information, sensing, and lasing applications. Such chirality must emanate from either the involved optical resonators or the quantum emitters. Here, we theoretically demonstrate a rare opportunity for realizing polaritons with so-called 2D chirality by strong coupling of the optical modes of (high finesse) achiral Fabry–Pérot cavities with samples exhibiting "apparent circular dichroism" (ACD). ACD is a phenomenon resulting from an interference between linear birefringence and dichroic interactions. By introducing a quantum electrodynamical theory of ACD, we identify the design rules based on which 2D chiral polaritons can be produced, and their chirality can be optimized.

Addressing and storing chirality in the optical fields of cavities and other resonators is of fundamental and technological importance. This issue has gained urgency with the rapid developments in the areas of quantum computing, quantum sensing, and quantum communication, prompting interest in photons as candidate quantum information carriers[1–5]. Photons are both highly mobile and weakly interacting, as a result of which their quantum state can be transported over long distances, while quantum information can be conveniently stored in their internal spin degree of freedom. Importantly, the manifestation of photonic spin as circularly-polarized light allows this information to be transduced to matter by means of chiroptical interactions, allowing photons to be straightforwardly incorporated in quantum networks[6–11]. Optical resonators provide a viable means to amplify the intrinsically-weak light–matter interactions, allowing the strong coupling regime to be reached[12]. Within this regime, photons hybridize with excitations of the involved material (quantum emitter), producing polaritons[13–18]. Chiral polaritons, where strong coupling is combined with high degrees of chiral dissymmetry, would offer the ideal conditions for photon-to-matter quantum transduction. Moreover, it would be of interest to chiral sensing[19,20] and provide an opportunity to realize chiral lasing.

While previous studies of chiral strong coupling have focused on plasmonics[21–26], such implementations suffer from considerable ohmic losses that inhibit the functionalities offered by polaritons. In that

regard, high finesse Fabry–Pérot (FP) cavities[27] make for a preferred optical resonator. Accordingly, the chiral symmetry should be broken for either the cavity or the quantum emitter (or both). The former is an interesting line of inquiry[28–36], but involves the challenge of simultaneously optimizing for chiral dissymmetry and for the finesse. Breaking the chiral symmetry in quantum emitters, on the other hand, is subject to a rather restrictive constraint imposed by (ordinary) FP cavities, namely that chiral dissymmetry is inverted for counter-propagating light. The latter is due to circular polarization switching handedness upon reflection at a mirror (i.e., switching from left- to right-handed and vice versa), as depicted in Fig. 1e[37]. As such, the quantum emitter must invert its handedness upon plane reflections parallel to the light propagation direction; a phenomenon referred to as two-dimensional (2D) chirality[38,39]. Optically-active samples[40] instead exhibit three-dimensional (3D) chirality associated with point reflections, thereby undergoing identical chiroptical interactions with counter-propagating light, as a result of which they do not produce chiral polaritons in a FP cavity. Perhaps due to these constraints, chiral polaritons based on FP cavities have to our knowledge not yet been experimentally realized.

In this paper, we demonstrate a rare opportunity for realizing 2D chiral polaritons in FP cavities based on quantum emitters exhibiting "apparent circular dichroism" (ACD). Not unlike FP cavities themselves, ACD as a scientific phenomenon traces back many decades[41],

[1]Department of Chemistry, Northwestern University, 2145 Sheridan Road, Evanston, IL 60208, USA. [2]Department of Chemistry, University of Wisconsin-Madison, Madison, WI 53706-1322, USA. ✉e-mail: roel.tempelaar@northwestern.edu

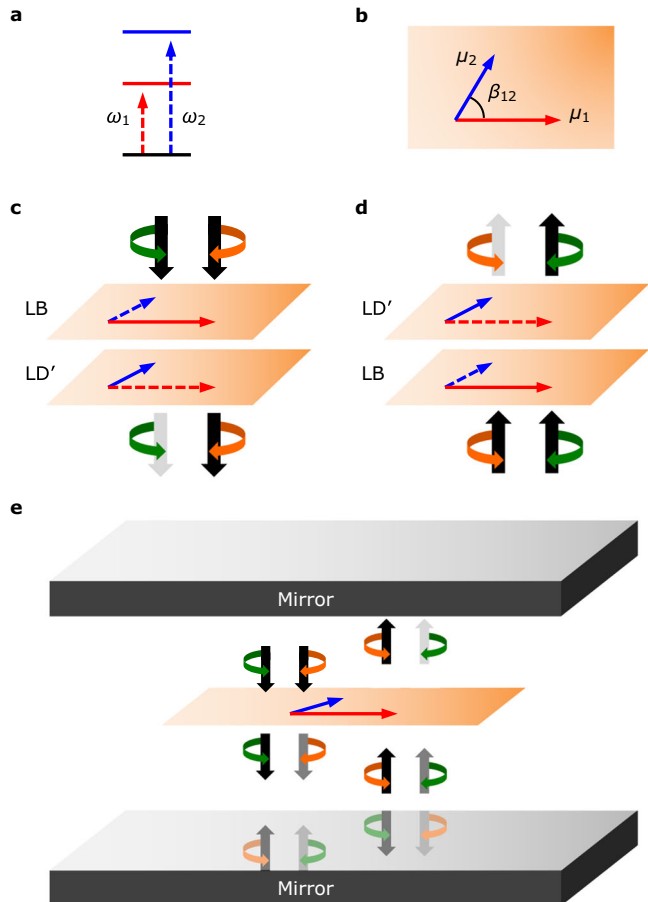

**Fig. 1 | Schematic depiction of apparent circular dichroism (ACD) and its implementation within an achiral Fabry–Pérot (FP) cavity.** Energy level diagram (**a**) and schematic of the transition dipoles (**b**) of a minimal ACD sample involving two bright, nondegenerate, and oblique transition dipoles. Angular transition frequencies $\omega_1$ and $\omega_2$, transition dipoles $\mu_1$ and $\mu_2$, and inter-dipole angle $\beta_{12}$ are indicated. Shown below is a schematic depiction of ACD due to linear birefringence (LB) and linear dichroism (LD) resulting from forward (**c**) and backward (**d**) propagation, yielding absorption of left-handed (green) and right-handed (orange) circularly-polarized light, respectively. The prime on LD indicates a 45° rotation in the plane of polarization. Also shown is a depiction of the interactions between an ACD sample and an achiral FP cavity (**e**) involving the selective absorption of the $\lambda = +$ polarization mode.

but in recent years has seen a marked increase in interest[42,43]. It results from oblique dichroic and birefringent axes of a sample, giving rise to a net difference in the absorption of clockwise and counter-clockwise rotating optical fields [corresponding to a differential absorption of left-handed and right-handed circularly-polarized light for a given irradiation direction[41,44] as depicted in Fig. 1c]. Within Mueller calculus, where light–matter interactions are described by matrices that manipulate polarization vectors[45], the leading contribution to ACD is given by

$$\text{ACD}(\omega) = \frac{1}{2}\left(\text{LD}(\omega) \cdot \text{LB}'(\omega) - \text{LD}'(\omega) \cdot \text{LB}(\omega)\right). \quad (1)$$

Here, LD and LB represent linear dichroism and linear birefringence, respectively, and the prime indicates a 45° axis rotation in the plane of polarization. ACD in principle fulfills the 2D chirality requisite for inverted chiroptical response, as shown in Fig. 1d. Moreover, ACD is of second (or higher) order in terms of optical interactions, such that its magnitude increases with sample thickness. As a result, (chiral) dissymmetry factors typically exceed those commonly found for optical

activity[46]. This suggests ACD to provide a unique opportunity for producing 2D chiral polaritons based on FP cavities.

In a recent work, we presented a microscopic yet semiclassical theory of ACD under weak light–matter coupling wherein the Mueller calculus treatment embodied by Eq. (1) was combined with a Lorentz oscillator model[47]. This allowed us to identify geometric sample properties that enable this process, namely the presence of oriented molecules featuring a pair of bright, nondegenerate, and oblique transition dipoles, as depicted in Fig. 1a, b. This work was motivated by a recent series of spectroscopic studies on ACD in organic thin films performed by Di Bari and coworkers[46,48–52]. In prior decades, ACD was mostly considered to be an optical artifact[53–56] arising especially in dye-doped cholesteric liquids[57–61], self-assembled fibers in solution[62,63], dyes bound in a linear orientation via a supporting matrix[64,65], and nanowires[63]. Indeed, ACD is not related to optical activity and is not to be confused with "real" CD, yet it may contaminate real CD measurements[53]. Perhaps for that reason, earlier theoretical works on ACD[66–70] restricted themselves to macroscopic Mueller or Jones[71] treatments, although there are some notable early efforts at connecting ACD to microscopic sample properties[41,72].

In demonstrating 2D chiral polaritons, the present Paper introduces a quantum electrodynamical theory of ACD. Departing from the semiclassical treatment from our previous work[47], which fails to describe strong coupling and polariton formation, the current theory is based on an appropriately-extended Jaynes–Cummings model[73], providing a quantum-mechanical treatment of a single quantum emitter in an ideal (lossless) cavity. It is shown that dissymmetries near the theoretical maximum can be achieved even within the single-molecule limit, as the FP cavity allows light to repeatedly interact with the quantum emitter, thereby increasing the optical path length. Generic rules are presented based on which the dissymmetry can be optimized, while particular attention is paid to the utilization of benzo[1,2–$b$:4,5-$b'$]dithiophene-based (BDT-based) oligothiophene as a 2D chiral quantum emitter.

## Results

### 2D chiral interaction terms

Our quantum electrodynamical theory of ACD is based on a suitably modified Jaynes–Cummings model[73]. Accordingly, the optical polarization is described using the natural basis of orthogonal clockwise and counter-clockwise rotating modes (for a given observer direction) inside an idealized Fabry–Pérot (FP) cavity[74], referred to as $\lambda = +$ and $\lambda = -$, respectively. The light–matter interaction Hamiltonian (within the rotating wave approximation) then takes the form

$$\hat{H}_{\text{int}} = i \sum_n \sum_{\lambda = \pm} A_{0,\lambda} \omega_n \tilde{\mu}_{n,\lambda} \left( \hat{a}_\lambda^\dagger \hat{b}_n - \hat{a}_\lambda \hat{b}_n^\dagger \right), \quad (2)$$

where $A_{0,\lambda}$ is the vector potential associated with the mode of polarization $\lambda$, and where $\hat{a}_\lambda^\dagger$ and $\hat{a}_\lambda$ are the associated photon creation and annihilation operators, respectively. If instead, the light–matter interaction Hamiltonian was expressed in terms of the more commonly-used $x$ and $y$ linearly polarized optical basis, our analysis would not change but the resulting expressions would be less intuitive.

In Eq. (2), $n$ runs over the excited states of the sample, with $\hat{b}_n^\dagger$ and $\hat{b}_n$ as the corresponding creation and annihilation operators, respectively. Moreover, $\hbar\omega_n$ is the associated excited state energy (where the ground state energy is taken to be zero as a reference), and $\tilde{\mu}_{n,\lambda}$ is the projection of the associated transition dipole vector onto the $\lambda$ polarization (which therefore obeys the same inversion antisymmetry). Notably, achiral as well as 3D chiral samples have $\tilde{\mu}_{n,+} = \tilde{\mu}_{n,-}$, as a result of which there is no selectivity with regard to the $\lambda = +$ and $\lambda = -$ polarization modes of the FP cavity. This selectivity will similarly vanish for a full three-dimensional orientational average of an ACD sample. Indeed, the ACD phenomenon relies on oriented

samples, although rotations in the plane normal to the light propagation direction do not diminish its effect[47].

It is important to emphasize that ACD (different from real CD) is not the result of a single chiroptical event at the microscopic level. Rather, it is a double-scattering event consisting of subsequent birefringent and dichroic interactions. Instead, the Hamiltonian given by Eq. (2) implicitly assumes a single absorptive event. Our approach to incorporate ACD in this Hamiltonian is, therefore, to only explicitly invoke the (second) dichroic interaction, and to include the (first) birefringent interaction through its effect on the transition dipole projection $\tilde{\mu}_{n,\lambda}$. In doing so, we first proceed to treat ACD semiclassically and within the regime of weak light–matter coupling. This treatment is agnostic to the presence or absence of a cavity, and merely considers ACD as a chiroptical response resulting from light with a given optical frequency propagating through a sample for a given optical path length $l$. Within this treatment, and assuming molecular crystals where intermolecular interactions are negligible, we have previously shown that the ACD transition rate takes the form[47]

$$ACD(\omega) = \frac{1}{2} l^2 \xi^2 \omega^2 \sum_{n,m} \mu_n^2 \sigma_n \mu_m^2 \omega_m V_n(\omega) W_m(\omega) \sin(2\beta_{mn}), \quad (3)$$

where $\omega$ is the optical (angular) frequency. Here, $n$ and $m$ run over the excited states of the involved molecule, with $\mu_n^2 = \tilde{\mu}_{n,+}^2 + \tilde{\mu}_{n,-}^2$ as the squared total dipole moment of state $n$, and $\beta_{mn}$ as the angle between the transition dipoles associated with states $n$ and $m$. It is assumed that all transition dipoles lie in the plane perpendicular to the light-propagation direction ($xy$ plane), or that any transition dipoles appearing in our analysis have been projected into this plane. In Eq. (3), $\xi \equiv (\hbar c v \epsilon_0 \sqrt{\epsilon_\infty})^{-1}$ with $v$ being the unit cell volume of the molecular crystal and $\epsilon_\infty$ being its effective high-frequency dielectric constant. Also appearing in Eq. (3) are the lineshape functions

$$\begin{aligned} W_n(\omega) &\equiv \frac{\omega_n^2 - \omega^2}{(\omega_n^2 - \omega^2)^2 + \gamma_n^2 \omega^2}, \\ V_n(\omega) &\equiv \frac{\gamma_n \omega}{(\omega_n^2 - \omega^2)^2 + \gamma_n^2 \omega^2}. \end{aligned} \quad (4)$$

These correspond to the real and imaginary frequency-dependent components of the dielectric susceptibility due to excited state $n$, respectively, and $\gamma_n$ is the associated damping parameter accounting for lineshape broadening. As can be verified by examination of Eq. (3), a minimal requisite for ACD signals incorporating the desired inversion antisymmetry is to have a minimum of two nondegenerate and nonparallel transition dipoles as well as macroscopic ordering[47].

Eq. (3) embodies a Fermi's Golden Rule treatment of ACD, similarly to that of mean (linear) absorption, the latter of which is given by

$$\bar{A}(\omega) = l \xi \omega \sum_n \mu_n^2 \omega_n V_n(\omega). \quad (5)$$

Alternatively, mean absorption can be expressed as $\bar{A}(\omega) = \frac{1}{2}(A_+(\omega) + A_-(\omega))$, with the $\lambda = \pm$ contributions given by

$$A_\lambda(\omega) \equiv 2l \xi \omega \sum_n \tilde{\mu}_{n,\lambda}^2 \omega_n V_n(\omega). \quad (6)$$

Key to establishing the form of $\tilde{\mu}_{n,\lambda}$ is that ACD can equivalently be expressed as $ACD(\omega) = \frac{1}{2}(A_+(\omega) - A_-(\omega))$. By comparison with Eq. (3), it then follows that

$$\tilde{\mu}_{n,\lambda} \equiv \mu_n \sqrt{\frac{1}{2} + \frac{1}{2} \tau_\lambda \sigma_n}, \quad (7)$$

where $\tau_\pm = \pm 1$ is an indexing variable, and $\sigma_n$ is the "2D chiral interaction term" defined as

$$\sigma_n \equiv \frac{1}{2} l \xi \omega \sum_{m \neq n} \mu_m^2 \omega_m W_m(\omega) \sin(2\beta_{mn}). \quad (8)$$

Here, it can be seen that for a given excited state $n$, other excited states project onto the 2D chiral interaction terms through their real dielectric dispersion, $W_m(\omega)$. As a result, $\tilde{\mu}_{n,\lambda}$ attains an $\omega$ dependence through $\sigma_n$. Importantly, the underlying birefringent interaction lends a linear dependence of $\sigma_n$ on $l$. While $\sigma_n$ should in principle be physically bounded by $-1$ and 1, corresponding to transition dipoles being entirely $-$ or $+$ polarized, respectively, its linear dependence on $l$ may violate this, leading to unphysical values of $\sigma_n$ as detailed below. The latter is a consequence of the second-order Mueller calculus treatment applied in order to arrive at Eq. (3).

Having established expressions for $\tilde{\mu}_{n,+}$ and $\tilde{\mu}_{n,-}$, we will proceed to consider a quantum electrodynamical model of ACD tailored to FP cavities. Accordingly, we replace the optical frequency $\omega$ by the resonance frequency of the FP cavity $\Omega$ set by the mode volume. Here, we will limit ourselves to the lowest-frequency resonance. Furthermore, $l$ now denotes the total path length due to repeated passes through the sample under internal cavity reflection. It should be noted that the Mueller calculus treatment underlying our ACD formalism implicitly considers the cavity mirrors to represent reciprocal boundaries imposed on the sample (see Methods). In practice, the path length will be limited by the cavity finesse. In this work we will ignore this effect, reserving its inclusion to a follow-up study. Instead, we loosely define the path length as the distance required for the average electromagnetic intensity to decay to $e^{-1}$ of its original magnitude when (repeatedly) propagating through the sample, meaning the value at which $\bar{A}(\Omega) = 1$ (assuming Arrhenius-type isotropic absorption). This yields

$$l = \frac{1}{\Omega \xi \sum_n \mu_n^2 \omega_n V_n(\Omega)}, \quad (9)$$

where we note that $l$ attains a $\Omega$ dependence due to the dispersion of the sample absorption. Substituting Eq. (9) into Eq. (8) then yields

$$\sigma_n = \frac{1}{2} \frac{\sum_{m \neq n} \mu_m^2 \omega_m W_m(\Omega) \sin(2\beta_{mn})}{\sum_m \mu_m^2 \omega_m V_m(\Omega)}. \quad (10)$$

Interestingly, this form of $\sigma_n$ is independent of the path length.

It is instructive to consider $\sigma_n$ for the case of two transition dipoles, which is the minimal configuration giving rise to finite ACD. In this case, Eq. (10) simplifies to

$$\begin{aligned} \sigma_1 &= \frac{1}{2} \frac{W_2(\Omega)/V_2(\Omega)}{1 + \Gamma_V \Gamma_{\mu^2} \Gamma_\omega} \sin(2\beta_{21}), \\ \sigma_2 &= -\frac{1}{2} \frac{W_1(\Omega)/V_1(\Omega)}{1 + \Gamma_V^{-1} \Gamma_{\mu^2}^{-1} \Gamma_\omega^{-1}} \sin(2\beta_{21}), \end{aligned} \quad (11)$$

where

$$\Gamma_V \equiv \frac{V_1(\Omega)}{V_2(\Omega)}, \quad \Gamma_{\mu^2} \equiv \frac{\mu_1^2}{\mu_2^2}, \quad \Gamma_\omega \equiv \frac{\omega_1}{\omega_2}. \quad (12)$$

We furthermore have that the numerators in Eq. (11) can be simplified as $W_n(\Omega)/V_n(\Omega) = (\omega_n^2 - \Omega^2)/(\gamma_n \Omega)$. Eq. (11) elucidates how different parameters impact the 2D chiral interaction terms. Specifically, their signs are governed by the sign of $\beta_{21}$, while both their magnitudes are maximized when $\beta_{21} = \pm 45°$, which was previously recognized to be the angle of maximum ACD[47,48,62]. Interestingly, $|\sigma_1|$ and $|\sigma_2|$ are oppositely impacted by $\Gamma_V$, $\Gamma_{\mu^2}$, and $\Gamma_\omega$, as increasing their values minimizes $|\sigma_1|$

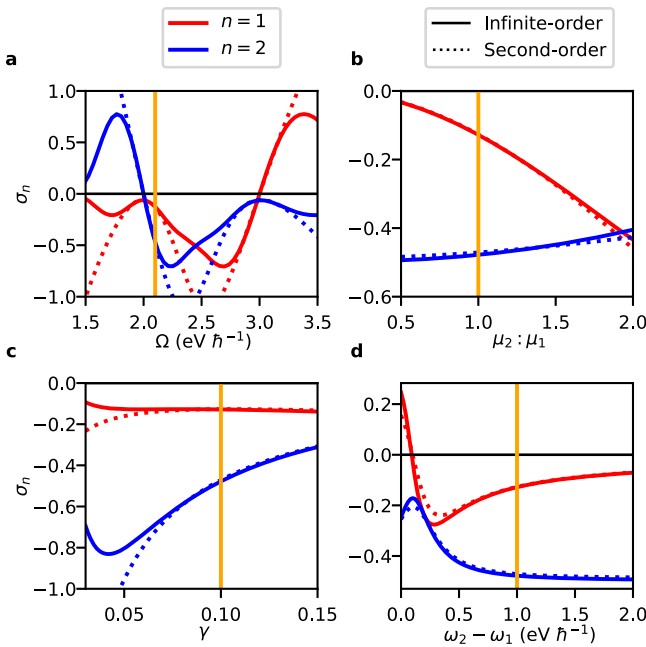

**Fig. 2 | Survey of 2D chiral interaction terms.** 2D chiral interaction terms $\sigma_n$ for two transition dipoles as a function of cavity frequency $\Omega$ (**a**), transition dipole moment ratio $\mu_2 : \mu_1$ with $\mu_1$ fixed (**b**), damping parameter $\gamma$ (**c**), and transition frequency gap $\omega_2 - \omega_1$ with $\omega_1$ fixed (**d**). Shown are results from a second-order Mueller calculus treatment, using Eq. (11) (dotted curve), alongside those from an infinite-order treatment (solid curve; see text), with $\omega_1 = 2.0$ eV $\hbar^{-1}$ and $\omega_2 = 3.0$ eV $\hbar^{-1}$ [except for (**d**)], while the angle between transition dipoles is set to $\beta_{21} = 45°$. Unless otherwise noted, $\gamma = 0.10$, $\mu_2 : \mu_1 = 1.0$, and $\Omega = 2.1$ eV $\hbar^{-1}$ in order to be slightly off-resonant with $\omega_1$. Other material parameters are set to $\mu_1 = 10.0$ D, $\upsilon = 4.5$ nm$^3$, and $\epsilon_\infty = 8.0$, with $\upsilon$ being the unit cell volume of the involved crystal and with $\epsilon_\infty$ being its effective high-frequency dielectric constant. The orange vertical line indicates points with the same parametrization across the different panels.

while it maximizes $|\sigma_2|$. It is also noteworthy that $W_n(\Omega) = 0$ at $\Omega = \omega_n$, as a result of which $\sigma_1 = 0$ at $\Omega = \omega_2$ and $\sigma_2 = 0$ at $\Omega = \omega_1$.

All of the trends discussed in the above are captured in Fig. 2, which shows $\sigma_1$ and $\sigma_2$ obtained through Eq. (11) under sweeps of the various parameters. Here, we have chosen the damping parameter to be proportional to $\omega_n$, such that $\gamma_n = \gamma\omega_n$ for constant $\gamma$, in order to account for the lifetimes of higher-lying states being typically shorter. From Fig. 2a it can be seen that $\sigma_1(\Omega - \omega_1) \approx \sigma_2(\omega_2 - \Omega)$, which would be a rigorous equality if we assumed $\gamma_1 = \gamma_2$. Figure 2b reflects the aforementioned scaling of both 2D chiral interaction terms with $\Gamma_{\mu^2}$. In this and the remaining Panels the 2D chiral interaction terms are depicted at $\Omega \approx \omega_1$, for which $\Gamma_V$ increases with increasing frequency gap, $\omega_2 - \omega_1$, and with decreasing $\gamma$. For the overall $\gamma$ dependence of $|\sigma_1|$, this contribution counteracts that of $W_n(\Omega)/V_n(\Omega)$, rendering $|\sigma_1|$ nearly constant under variations of $\gamma$, as demonstrated in Fig. 2c. For $|\sigma_2|$, on the other hand, both contributions act constructively as a result of which this term decreases with $\gamma$. Lastly, Fig. 2d depicts how $|\sigma_1|$ decreases with increasing $\omega_2 - \omega_1$ (through an increase in $\Gamma_V$) with the reverse dependence apparent for $|\sigma_2|$. These trends break down near $\omega_2 - \omega_1 = 0$ at which point we instead find that $\sigma_2 = -\sigma_1$, since all parameters appearing in Eq. (11) are identical. We note that whereas the 2D chiral interaction terms are proportional to ACD normalized to total absorbance, we previously analyzed the dependence of ACD itself on the frequency gap, and found that for ACD to be appreciable the involved transitions should be within each other's linewidth[47].

From Fig. 2a it becomes obvious that Eq. (11) may yield 2D chiral interaction terms that break the physical bounds of −1 and 1. This is particularly so for $\Omega$ off resonance with $\omega_1$ and $\omega_2$, where an increasing

path length $l$ causes a failure of the underlying second-order Mueller calculus treatment, as previously mentioned. This failure can be remedied by replacing the second-order Mueller calculus treatment by its infinite-order variant, following previous work by Brown[75,76]. Accordingly, $\frac{1}{2}l$ in Eq. (8) is replaced by an oscillatory function $B_1(l)/l$ (see Methods). Within this treatment, assessing the path length needs the $e^{-1}$ decay distance to be evaluated numerically. Results for $\sigma_n$ within this improved treatment are shown alongside those from Eq. (11) in Fig. 2, and are indeed found to obey the physical bounds. Notably, for $\Omega$ in resonance with $\omega_1$ and $\omega_2$, results from Eq. (11) are seen to be in close agreement with the infinite-order treatment. Importantly, even when physically bounded, the 2D chiral interaction terms are seen to approach $\pm 1$ for reasonable parameter values, meaning that levels of 2D chirality near the theoretical maximum can be achieved for ACD under ideal FP cavity confinement. We note that $\epsilon_\infty$ and $\upsilon$ have no impact on the second-order treatment, but that they do have a (minor) effect on the infinite-order 2D chiral interaction terms, which we employ for the remainder of this Paper.

## 2D chiral polaritons

We will proceed to evaluate the polaritonic states arising from ACD samples embedded in an idealized achiral FP cavity. Assuming the light propagation direction to be normal to the cavity plane at all times, the relevant optical modes organize in degenerate pairs with orthogonal polarization. Such cavity by itself does not break the symmetry between the two modes, and as such modes within any orthogonal polarization basis will form valid intrinsic modes of the cavity. Embedding of an ACD sample breaks this symmetry within the basis of $\lambda = +$ and $\lambda = -$ polarized modes. To describe this, we extend the light–matter interaction Hamiltonian from Eq. (2) within the single-molecule limit in order to include the diagonal photonic and molecular excitation energies, yielding the total Hamiltonian

$$\hat{H} = \hbar\Omega \sum_{\lambda = \pm} \hat{a}_\lambda^\dagger \hat{a}_\lambda + \sum_n \hbar\omega_n \hat{b}_n^\dagger \hat{b}_n$$
$$+ i \sum_n \sum_{\lambda = \pm} A_{0,\lambda} \omega_n \mu_n \sqrt{\frac{1}{2} + \frac{1}{2}\tau_\lambda \sigma_n} \left( \hat{a}_\lambda^\dagger \hat{b}_n - \hat{a}_\lambda \hat{b}_n^\dagger \right). \quad (13)$$

Here, we introduced the 2D chiral interaction terms through Eq. (7). The polaritonic eigenstates of this Hamiltonian follow from the time-independent Schrödinger equation, $\hat{H}|\Psi^\alpha\rangle = E^\alpha|\Psi^\alpha\rangle$. Within the manifold of single excitations (meaning a single photon or molecular excited state) the eigenstates take the general form

$$|\Psi^\alpha\rangle = C_e^\alpha |\psi_e^\alpha\rangle + C_\gamma^\alpha |\psi_\gamma^\alpha\rangle,$$
$$|\psi_e^\alpha\rangle = \sum_n d_n^\alpha |n\rangle, \quad (14)$$
$$|\psi_\gamma^\alpha\rangle = \sum_\lambda d_\lambda^\alpha |\lambda\rangle,$$

where we first applied an expansion into the total contributions from the molecular excited states (denoted "e") and those from the optical modes (denoted $\gamma$), effectively invoking Hopfield coefficients[77], followed by sub-expansions of each. In the above, $|n\rangle \equiv \hat{b}_n^\dagger|0\rangle$ and $|\lambda\rangle \equiv \hat{a}_\lambda^\dagger|0\rangle$, with $|0\rangle$ representing the vacuum state without molecular or optical excitations.

In order to characterize the polaritonic eigenstates it proves convenient to define a set of scalar metrics, the first of which is given by

$$g^\alpha \equiv 2 \frac{|d_+^\alpha|^2 - |d_-^\alpha|^2}{|d_+^\alpha|^2 + |d_-^\alpha|^2}. \quad (15)$$

This metric is analogous to the dissymmetry factor commonly used to characterize chiroptical signals, $g \equiv 2(A_+ - A_-)/(A_+ + A_-)$, and quantifies the anisotropy in the admixture of both optical polarizations into the polariton. Note that $g^\alpha$ is bounded as $-2 \leq g^\alpha \leq 2$. A second metric is introduced in order to quantify the polaritonic mixing,

$$\chi^\alpha \equiv 2|C_e^\alpha C_\gamma^\alpha|. \tag{16}$$

This metric is maximized for eigenstates evenly split between molecular and photonic excitations, and is bounded as $0 \leq \chi^\alpha \leq 1$. While $g^\alpha$ and $\chi^\alpha$ quantify the degree of 2D chirality and light–matter hybridization, respectively, combining them yields a single scalar metric quantifying 2D chiral light–matter hybridization. Accordingly, the "polaritonic dissymmetry factor" is defined as

$$\tilde{g}^\alpha \equiv g^\alpha \chi^\alpha, \tag{17}$$

which is bounded as $-2 \leq \tilde{g}^\alpha \leq 2$. Maximal $|\tilde{g}^\alpha|$ implies both optimal polaritonic mixing and optimal 2D chirality, whereas a vanishing $\tilde{g}^\alpha$ implies that either polaritonic mixing or 2D chirality (or both) is absent. It should be noted that any sign changes in $\tilde{g}^\alpha$ are due to $g^\alpha$, since $\chi^\alpha > 0$ by construction.

Shown in Fig. 3 are polariton dispersions, combined with a representation of the polaritonic dissymmetry $\tilde{g}^\alpha$ as a function of the cavity frequency $\Omega$, assuming an achiral FP cavity by setting $A_{0,+} = A_{0,-} = A_0/\sqrt{2}$, such that the total vector potential obeys the Pythagorean equality $A_0^2 = A_{0,+}^2 + A_{0,-}^2$ between orthogonal modes. These results were obtained by solving the time-independent

Schrödinger equation through numerical diagonalization of the total Hamiltonian given by Eq. (13) while representing the molecule by the minimal configuration of two transition dipoles with $\mu_1 = \mu_2$ and $\beta_{21} = 45°$. Combined with the two orthogonal photonic states, this yields a total of four eigenstates. These states are depicted in Fig. 3 for two values of $A_0$ (the largest of which being still amenable to the rotating-wave approximation). The dispersions shown in Fig. 3 exhibit the known behavior of achiral polaritons, including a Rabi splitting in the regions where $\Omega$ crosses the excited state transitions. Unsurprisingly, this Rabi splitting is seen to increase with $A_0$, replicating the behavior of an achiral Jaynes–Cummings model[73]. Importantly, however, there is an undispersed state following the light line in each crossing region.

In Fig. 3, $\tilde{g}^\alpha$ is seen to assume a bisignate profile that changes minimally with increasing $A_0$, apart from an overall increase in amplitude. This suggests that the $A_0$ dependence is primarily confined to the polaritonic mixing $\chi_\alpha$ in Eq. (17), while the (bare) dissymmetry $g^\alpha$ is largely insensitive to $A_0$. Importantly, the polaritonic dissymmetry reaches values of $\tilde{g}^\alpha \sim 1.0$, which is a substantial fraction of the theoretically-maximum value. This is particularly remarkable since we are considering a single molecule which within a conventional Mueller calculus treatment of absorption would not have an ACD response (this response being a second-order effect at the minimum) but which is allowed to strongly and repeatedly interact with itself within the FP cavity. Within the resulting sequence of interactions the 2D chirality continuously increases; an effect that is limited by the optical path length, $l$.

Figure 4 systematically explores the behavior of $\tilde{g}^\alpha$ for the lowest-energy polaritonic eigenstate, $\alpha = 1$, which is expected to be the most thermodynamically stable and therefore the most likely candidate for

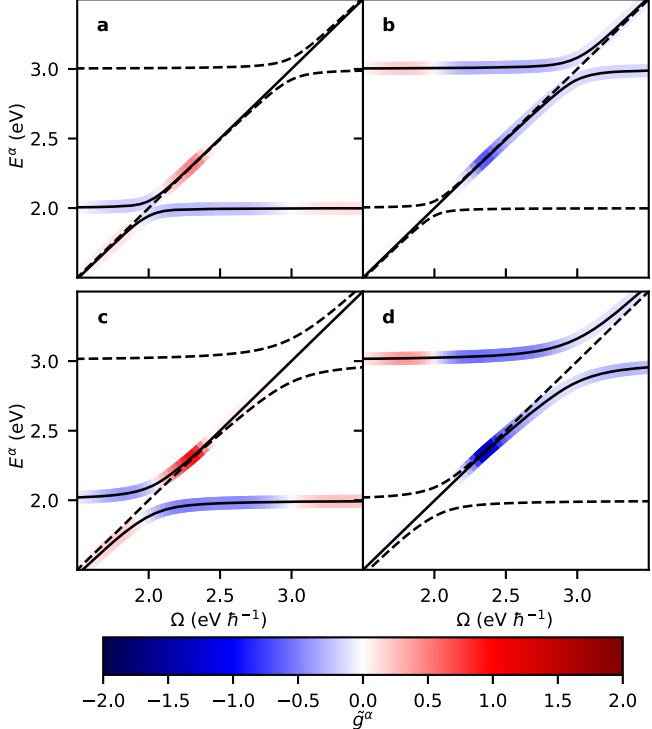

**Fig. 3 | Polaritonic dissymmetry $g^\alpha$ as a function of (angular) cavity frequency $\Omega$ for the minimal configuration of two transition dipoles.** Shown are results for a vector potential $A_0 = 35$ eV $e^{-1}c^{-1}$ (**a**, **b**) and for $A_0 = 70$ eV $e^{-1}c^{-1}$ (**c**, **d**). Depictions of $\tilde{g}^\alpha$ are separated over two Panels to avoid overlap, and in each Panel corresponds to the polariton dispersions indicated by the solid curves (other polariton dispersions are indicated by the dashed curves as a reference). Results are obtained by numerical diagonalization of the Hamiltonian given by Eq. (13), with inter-dipole angle $\beta_{21} = 45°$, (angular) transition frequencies $\omega_1 = 2.0$ eV $\hbar^{-1}$ and $\omega_2 = 3.0$ eV $\hbar^{-1}$, transition dipole moments $\mu_1 = \mu_2 = 10.0$ D, crystal unit cell volume $\nu = 4.5$ nm$^3$, and high-frequency dielectric constant $\epsilon_\infty = 8.0$.

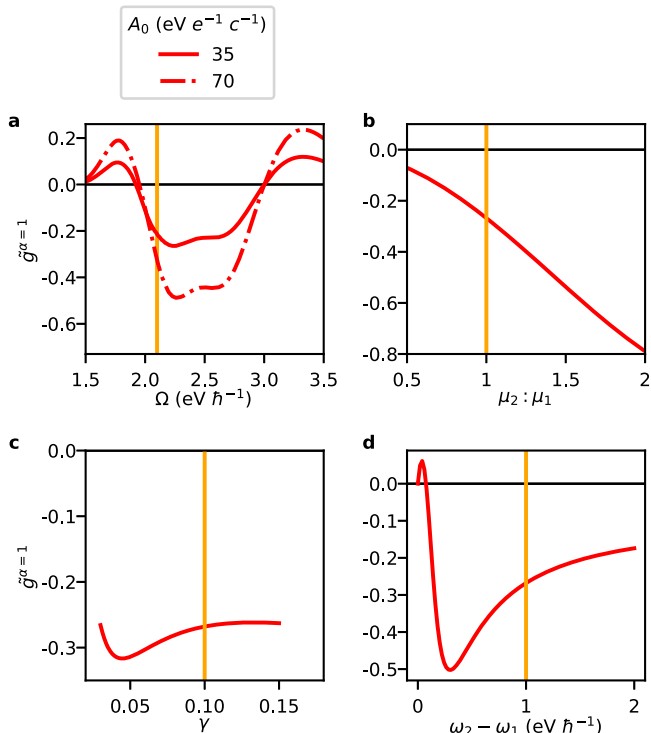

**Fig. 4 | Polaritonic dissymmetry for the lowest-energy polariton, $g^{\alpha=1}$.** Shown are results as a function of (angular) cavity frequency $\Omega$ (**a**), transition dipole moment ratio $\mu_2 : \mu_1$ (**b**), damping factor $\gamma$ (**c**) and (angular) transition frequency difference $\omega_2 - \omega_1$ (**d**). Unless noted otherwise, $A_0 = 35$ eV $e^{-1}c^{-1}$ and all other parameters are identical to Fig. 2, including an inter-dipole angle $\beta_{21} = 45°$, a crystal unit cell volume $\nu = 4.5$ nm$^3$, and an effective high-frequency dielectric constant $\epsilon_\infty = 8.0$. The orange vertical line indicates points with the same parametrization across the different panels.

steady-state chiroptical applications. Shown in Fig. 4a is the $\Omega$ dependence of $\tilde{g}^{\alpha=1}$, emphasizing the bisignate profile previously observed in Fig. 3, and further confirming that an increase in $A_0$ primarily acts as an overall rescaling of this quantity governed through $\chi^{\alpha=1}$. Based on Fig. 4b–d it is tempting to assert that the dependence of $\tilde{g}^{\alpha=1}$ on $\mu_2 : \mu_1$, $\gamma$, and $\omega_2 - \omega_1$ is instead governed by the bare dissymmetry $g^{\alpha=1}$, seeing the strong similarities with the trends depicted in Fig. 2b–d, but further analysis is necessary in order to substantiate this.

### Three-state approximation

To better understand the computational results shown in Figs. 3 and 4, we will proceed with a purely-analytical treatment of the total Hamiltonian given by Eq. (13). This Hamiltonian generally assumes a dimensionality of $(N+2) \times (N+2)$ for a total of $N$ excited states of the sample. Hence, for the minimal configuration of two transition dipoles, one would have to analytically diagonalize a $4 \times 4$ matrix, which is generally not feasible. However, to an approximation, it is possible to describe the polaritons resulting from Eq. (13) by restricting the explicit Hilbert space to a single excited state $(n)$, while including the other excited states through their contribution to the 2D chiral interaction term $\sigma_n$. This would be a good approximation provided that excited state $n$ mixes more strongly with the photons than all other states due to having larger coupling elements or due to its transition frequency being closer in resonance with the cavity frequency, $\omega_n \approx \Omega$. Within this "three-state approximation" (TSA) the time-independent Schrödinger equation takes the form

$$
\begin{pmatrix}
\hbar\Omega & 0 & \Phi_n\sqrt{\frac{1}{2}+\frac{1}{2}\sigma_n} \\
0 & \hbar\Omega & \Phi_n\sqrt{\frac{1}{2}-\frac{1}{2}\sigma_n} \\
\Phi_n\sqrt{\frac{1}{2}+\frac{1}{2}\sigma_n} & \Phi_n\sqrt{\frac{1}{2}-\frac{1}{2}\sigma_n} & \hbar\omega_n
\end{pmatrix}
\left|\Psi^\alpha_{(n)}\right\rangle = E^\alpha_{(n)}\left|\Psi^\alpha_{(n)}\right\rangle,
\tag{18}
$$

where $\Phi_n \equiv |A_0|\omega_n\mu_n$ is the achiral light–matter interaction strength, taken here to be purely real without loss of generality, and where the subscript $(n)$ emphasizes the sample excited state for which the TSA is taken.

The eigenvalue equation given by Eq. (18) is analytically solvable due to the sparsity of the $3 \times 3$ Hamiltonian matrix. Three solutions are found, one of which consists of purely-photonic contributions,

$$
\left|\Psi^\gamma_{(n)}\right\rangle =
\begin{pmatrix}
-\sqrt{\frac{1}{2}-\frac{1}{2}\sigma_n} \\
\sqrt{\frac{1}{2}+\frac{1}{2}\sigma_n} \\
0
\end{pmatrix},
\tag{19}
$$

with an associated eigenenergy $E^\gamma_{(n)} = \hbar\Omega$. This explains the undispersed state observed at each crossing in Fig. 3. The other two solutions constitute an upper (u) and lower (l) polariton branch, and are given by

$$
\left|\Psi^{\mathrm{u/l}}_{(n)}\right\rangle = \frac{1}{\sqrt{\left(\Pi^{\mathrm{u/l}}_{(n)}\right)^2+1}}
\begin{pmatrix}
\Pi^{\mathrm{u/l}}_{(n)}\sqrt{\frac{1}{2}+\frac{1}{2}\sigma_n} \\
\Pi^{\mathrm{u/l}}_{(n)}\sqrt{\frac{1}{2}-\frac{1}{2}\sigma_n} \\
1
\end{pmatrix},
\tag{20}
$$

with

$$
\Pi^{\mathrm{u/l}}_{(n)} \equiv \frac{\Phi_n}{\Delta_n \pm \sqrt{\Delta_n^2+\Phi_n^2}},
\tag{21}
$$

and where $\Delta_n \equiv 2\hbar(\omega_n - \Omega)$ is twice the energetic detuning. The corresponding eigenenergies are given by

$$
E^{\mathrm{u/l}}_{(n)} = \hbar\frac{\Omega+\omega_n}{2} \pm \sqrt{\Delta_n^2+\Phi_n^2}.
\tag{22}
$$

Substituting the above eigensolutions into the (bare) dissymmetry factor defined in Eq. (15) yields (see Supplementary Methods 1 for details)

$$
g^{\mathrm{u/l}}_{(n)} = 2\sigma_n.
\tag{23}
$$

Interestingly, within the second-order Mueller calculus treatment this dissymmetry factor is independent of $A_0$, cf. Eq. (10), with a (weak) dependence only being possible through higher-order effects contained in the oscillatory function $B_1(l)/l$. This substantiates our observations in Figs. 3 and 4 that the effect of $A_0$ is largely manifested in the polaritonic mixing. Within the TSA, this mixing is obtained by substituting the above eigensolutions in Eq. (16), yielding (see Supplementary Methods 1)

$$
\chi^{\mathrm{u/l}}_{(n)} = \frac{\Phi_n}{\sqrt{\Delta_n^2+\Phi_n^2}}.
\tag{24}
$$

As expected, this mixing is bounded by 1, corresponding to a perfect split between photonic and electronic states at resonance, $\Delta_n = 0$, and decreases with increasing $\Delta_n$ or decreasing $\Phi_n$.

In order to assess the accuracy of the TSA, we compare in Fig. 5 TSA results for $n = 1$ against the numerical solutions of the full Hilbert space for the $A_0 = 35$ eV $e^{-1}c^{-1}$ case previously considered in Fig. 3. Shown as a function of $\Omega$ are the polariton dispersions and the polaritonic dissymmetry factors. The dispersions are indistinguishable from those predicted by the full Hilbert space, indicating that mixing due to $n = 2$ has a negligible impact on $E^{\mathrm{u/l}}_{(n=1)}$. In contrast, discrepancies are observed for the polaritonic dissymmetry factors. Within the TSA the upper and lower polariton pair have identical 2D chirality, $g^{\mathrm{u}}_{(n=1)} = g^{\mathrm{l}}_{(n=1)}$, while the overall polaritonic dissymmetry is seen to be monosignate. Within the full Hilbert space, $g^{\mathrm{u}}_{(n=1)} = g^{\mathrm{l}}_{(n=1)}$ is violated for polaritonic states close to the light line (for which $E^{\mathrm{u/l}}_{(n=1)} \approx \Omega$), giving rise

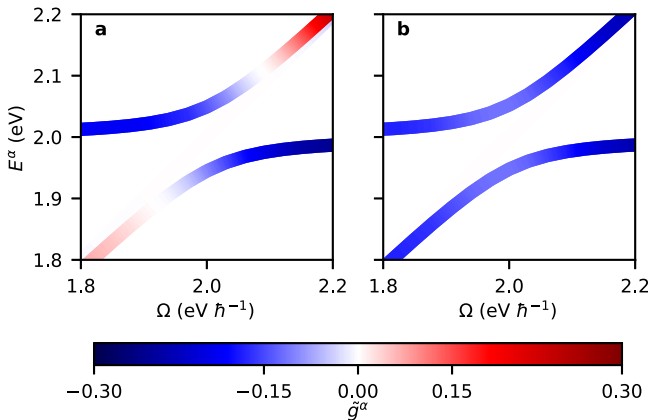

**Fig. 5 | Polaritonic dissymmetry and dispersion with and without the three-state approximation (TSA).** Polaritonic dissymmetry $\tilde{g}^\alpha$ as a function of (angular) cavity frequency $\Omega$ for the minimal configuration of two transition dipoles obtained through a numerical diagonalization of the Hamiltonian given by Eq. (13) (**a**), and comparative results within the TSA (**b**) which only includes explicitly the lowest-energy excited state ($n = 1$). Unless noted otherwise, parameters are identical to Fig. 2, while the vector potential is taken to be $A_0 = 35$ eV $e^{-1}c^{-1}$. Note that colors have been linearly interpolated for a smoother gradient.

to bisignate dissymmetry factors. Discrepancies between the TSA and the full Hilbert space results are rationalized by the two excited states of the sample coupling only indirectly to one another through the photonic states. When separated from the light line, polaritonic states have a substantial matter contribution due to a single resonant excited state, and the other excited state mixes only weakly through the photonic component. As a result, $g_{(n)}^{u/l} = 2\sigma_n$ governs the polaritonic dissymmetry, rendering this relationship a useful (albeit approximate) manner with which one can tune 2D chirality. Close to the light line, however, polaritonic states instead contain a predominant photonic component that couples more equally to both excited states (both being offresonant), as a result of which $g_{(n)}^{u/l} \neq 2\sigma_n$, rendering the TSA inaccurate.

With the analytical insights offered by the TSA we now revisit Fig. 4. For the lowest-energy polaritonic state depicted here, the regime of validity of the TSA is where $\Omega > \omega_1$ (away from the light line) while $\Omega$ still being sufficiently separated from $\omega_2$. Within this regime, $\tilde{g}^{\alpha=1} \approx \chi^{\alpha=1}\sigma_1$, and the profile of $\tilde{g}^{\alpha=1}$ as a function of $\Omega$ can be rationalized based on the dispersion of $\sigma_1$ in Fig. 2a, while appreciating that $\chi^{\alpha=1}$ monotonically decreases away from the resonance $\Omega \approx \omega_1$. We also note that the sign change of $\tilde{g}^{\alpha=1}$ at ~3 eV is due to a change of sign of $\sigma_1$. However, at ~2 eV, which lies outside the TSA regime, the sign change of $\tilde{g}^{\alpha=1}$ is instead due to a mixing of the $n=2$ excited state. The TSA analysis furthermore confirms that the $A_0$ dependence is confined to $\chi^{\alpha=1}$ whereas the $\mu_2 : \mu_1$, $\gamma$, and $\omega_2 - \omega_1$ dependence is confined to $g^{\alpha=1}$. In particular, for the results shown in Fig. 4b–d we have $\chi^{\alpha=1} \approx 1$ (due to $\Omega \approx \omega_1$) as a result of which $\tilde{g}^{\alpha=1} \approx 2\sigma_1$, which is readily verified through a comparison with Fig. 2b–d. We further note that in Fig. 3, $\sigma_1(\Omega - \omega_1) \approx \sigma_2(\omega_2 - \Omega)$ governs the behaviors of $\tilde{g}^{\alpha=1}$ and $\tilde{g}^{\alpha=2}$ in their respective regions away from the light line (which follows from an application of the TSA to both excited states). Lastly, it should be pointed out that the polaritonic dissymmetry undergoes a global sign change upon inverting the inter-dipole angle (see Supplementary Fig. 1). Such is evident from the analytical expressions within the TSA (seeing that it depends on the 2D chiral interaction terms, and cf. Eq. (11)), and can generally be understood by the principle that such angle inversion represents a sample flipping, which for 2D chiral samples returns their (2D) enantiomer.

## Design rules for 2D chiral polaritons

Through both our numerical and approximate analytical treatments of the quantum electrodynamical theory of ACD, we have arrived at a set of design rules for optimizing the 2D chirality of polaritons, embodied by $\tilde{g}^{\alpha}$. These design rules can be summarized as follows.

1. As with polaritonic states in general, the frequency of the cavity mode should be approximately resonant with that of some transition of the quantum emitter, $\Omega \approx \omega_n$ [see Eq. (24)].
2. Other quantum emitter transition frequencies ($\omega_m$) must be sufficiently close to $\Omega$ for the ACD interactions to be relevant, while being sufficiently separated from $\omega_n$ to have well-resolved polariton states [see Fig. 4d].
3. Dipole moments of those other transitions are preferably large compared to that of the resonant transition, $\mu_m \gg \mu_n$, while their mutual angle approaches 45° [see Eqs. (11) and (23)].
4. Lastly, there are considerations regarding energetic stability (not studied in the present work), which suggests that the resonant transition preferably involves the lowest-energy excited state of the quantum emitter, rendering the resulting polariton optimally stable against relaxation pathways.

The minimal configuration for generating 2D chiral polaritons based on ACD is that of a quantum emitter featuring two bright, nondegenerate, and nonparallel transitions. For such minimal configuration, the above design rules predict that polaritons reach the highest degree of 2D chirality once the weakest transition is approximately resonant with the cavity mode, whereas the strongest transition is energetically well-separated yet sufficiently close to the first, while the two transitions have a mutual orientation angle of 45°.

## Application to BDT-based oligothiophene

Having in place the design rules for optimizing 2D chiral polaritons based on ACD, we now turn our attention to BDT-based oligothiophene. Thin films composed of this molecule were the specific focus of our previous work introducing a microscopic yet semiclassical treatment of ACD under weak light–matter coupling[47]. Importantly, the intermolecular electronic interactions were found to be weak for these films[47], as a result of which our microscopic treatment was directly applicable. Excellent agreement was found for linear absorption and ACD spectra against experimental results[48] upon including three electronic ground-to-excited state transitions coupled to a high-frequency intramolecular vibration, and parametrized based on electronic structure calculations and spectral fitting. We will proceed to theoretically predict the 2D chiral polaritons that would arise when BDT-based oligothiophene serves as a quantum emitter in an achiral FP cavity. As in the previous Sections, we describe this setup within the limit of a single molecule through application of the modified Jaynes–Cummings model incorporating the quantum-electrodynamical theory of ACD.

Figure 6a shows the calculated polaritonic dissymmetry and dispersions. We refer to our previous work[47] for the applicable molecular parameters (and reiterate the procedure followed for determining

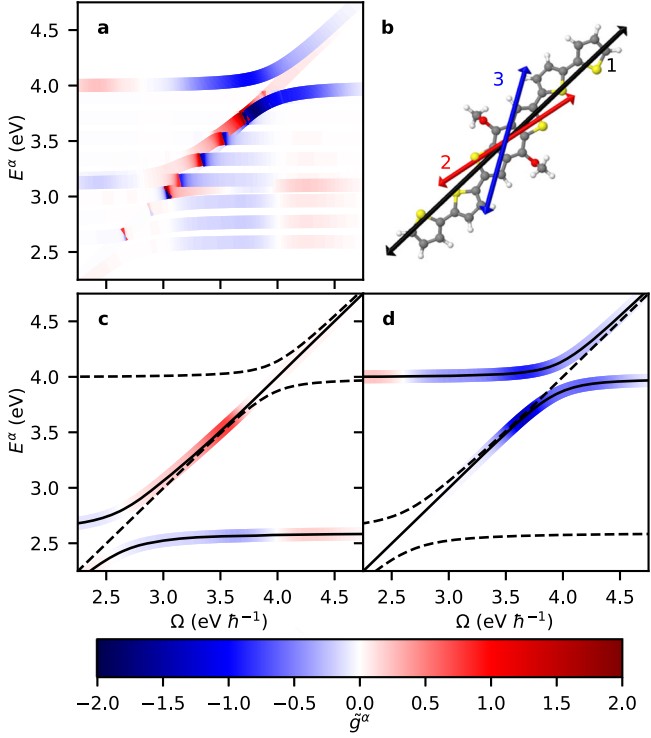

**Fig. 6 | 2D chiral polaritons in BDT-based oligothiophene.** Polaritonic dissymmetry $\tilde{g}^{\alpha}$ for BDT-based oligothiophene as predicted by our theory (**a**). Overlapping states drawn in order of $|\tilde{g}^{\alpha}|$ such that states with greatest $|\tilde{g}^{\alpha}|$ are visible. The molecular model of the BDT-based oligothiophene was adopted from our previous work[47] (see text for details). Molecular structure shown in (**b**) with relevant ground-to-excited state transition dipoles indicated, labeled 1, 2, and 3, in order of increasing transition energy (orientation of dipoles relative to the molecular structure is arbitrary to within variations between electronic structure calculations and spectral fitting[47]). Also shown are results without the vibrational modes and only including transitions 1 and 3 (**c**, **d**).

these parameters under Methods), but note that we adjusted the unit cell volume to $v = 4.46 \, nm^3$, in order to more adequately account for the oligothiophene side chains, and the high-frequency dielectric constant to $\epsilon_\infty = 8.0$, to better agree with relevant experimental high-frequency dielectric constants[78]. As previously discussed[47], ACD in BDT-based oligothiophene is due to interactions of a strong electronic transition at $\omega_1 = 2.6$ eV (with $\mu_1 = 13.7$ D) interfering with two weaker transitions at $\omega_2 = 3.1$ eV (with $\mu_2 = 6.7$ D) and $\omega_3 = 4.0$ eV (with $\mu_2 = 6.4$ D) under angles $\beta_{12} = -10.5°$ and $\beta_{13} = 31.5°$. The lowest transition is furthermore coupled to a vibrational mode with an energy of 185 meV. The resulting manifold of vibronic excited states gives rise to a large number of pairwise interactions between transitions that contribute to the 2D chiral interaction terms, and thus a large number of 2D chiral polaritons. This is borne out in Fig. 6a where a panoply of polariton resonances can be seen. What is immediately obvious is that very large polaritonic dissymmetries are reached, with values of $|\tilde{g}^\alpha|$ up to -1.8, which forms a stark contrast with bare BDT-based oligotiophene thin films that reach typical dissymmetry values of -0.02[48]. Again, the significant gain in dissymmetry upon polariton formation is due to the molecule being able to strongly and repeatedly interact with itself within the FP cavity.

In spite of the large number of states, the regions of large $|\tilde{g}^\alpha|$ can largely be understood based on a simplified model including the electronic transition at 4.0 eV interacting with the electronic transition at 2.6 eV (and without the inclusion of vibronic coupling). This is borne out in Fig. 6c, d where we reproduce the salient features shown in Fig. 6a by restricting the model to this pair of transitions. Both the dispersions and polaritonic dissymmetries resulting from this reduced model can in turn be understood based on our analysis of the two-dipole system in Fig. 3. Moreover, the involved transitions can be shown to satisfy the majority of the design rules presented in the previous Section, especially with the cavity frequency in resonance with the molecular transition at 4.0 eV. In that case, $\tilde{g}^\alpha$ benefits from the comparatively larger dipole moment of the transition at 2.6 eV, which adds to the favorable inter-dipole angle of 31.5°[47]. A potential pitfall would be that the resonant molecular transition is the highest in energy rather than the lowest, and a reversal of these excited state properties is likely to yield 2D chiral polaritons with an improved energetic stability. Regardless, these results offer encouraging prospects for the practical implementation of 2D chiral polaritons based on existing ACD samples.

## Discussion

In this paper, we have shown that a rare opportunity for generating 2D chiral polaritons is provided by embedding ACD samples in achiral FP cavities. We have predicted and characterized such polaritons based on a suitably-modified Jaynes–Cummings model incorporating a quantum electrodynamical theory of ACD, and shown that (chiral) dissymmetry factors approach their theoretical maximum values even when taking the quantum emitter in the single-molecule limit. This is due to ACD being of second or higher order in terms of the light–matter interaction, allowing exceptionally-high dissymmetry factors to be realized as the quantum emitter interacts with itself through the cavity. Moreover, the inverted chiroptical response originating from ACD proves compatible with achiral FP cavities. As such, 2D chiral polariton engineering efforts can optimize exclusively for cavity finesse, without having to additionally achieve chiral selectivity of the cavity. We applied our theory to BDT-based oligothiophene, which was previously experimentally[48] and theoretically[47] scrutinized for its pronounced ACD response, and provided indications that high dissymmetries are attainable for experimentally realizable samples.

Our future efforts will be directed towards understanding the role of cavity finesse as well as collective effects arising from multiple cavity-confined ACD quantum emitters. The finesse is expected to predominantly affect the effective path length, which in turn impacts the 2D chiral interaction terms through Eq. (8). On the other hand, an increasing number of quantum emitters increases the chiroptical interaction strength per cavity cycle. We a priori expect the results presented in the current manuscript to have minor sensitivity to the dark state manifold that is purported to act as a reservoir for polariton dynamics under collective strong coupling[18], because the 2D chiral discrimination manifests in experimentally-observed Rabi components, and since the relevant chiroptical phenomena manifest in the impulsive photoexcitations rather than in thermalized polaritonic states. However, for the purpose of using ACD samples to temporarily store 2D chiral polarizations, it will be important to understand the role of the dark state reservoir in directing the polarization evolution beyond the initial photoexcitation. For the photoexcitation event itself, perhaps a more pertinent issue is the possibility of helically-stacked molecules giving rise to a 3D form of ACD with non-inverted chiroptical selection rules[47]. This will inhibit the 2D polaritonic chirality, and will need to be suppressed in the synthesis of ACD samples. It will also be of interest to consider resonances between ACD samples and higher-energy modes of a cavity, and to assess whether polaritonic dissymmetry factors can be enhanced or suppressed through mode profiles.

Chiral polaritons based on FP cavities have to our knowledge yet to be experimentally realized. In addition to the challenge of engineering high-finesse chiral cavities and/or strongly-chiral quantum emitters, the constraint of 2D chirality imposed by FP cavities radically reduces engineering opportunities. Due to its 2D chiral properties and high chiral discrimination, we anticipate ACD to provide perhaps the most plausible route towards the experimental realization of cavity-based 2D chiral polaritons. Since the clockwise and counter-clockwise rotating cavity modes correspond to right- or left-handed circularly-polarized light traveling in a given direction, any leakage out of the cavity will be circularly polarized. Hence, an immediate application enabled by cavity-based 2D chiral polaritons is chiral lasing without the use of macroscopic optical components. This adds to the prospect of harnessing 2D chiral polaritons for the transduction between photonic spin and polarized excitations in matter. Here, we foresee that ACD samples can serve the role of the matter component, such as explored in the present study, but that it may also enable a symmetry breaking between 2D chiral excitations in monolayer transition-metal dichalcogenides[33] or porphyrin molecules[34] by simultaneous strong coupling with such materials on the one hand and ACD samples on the other, which is worthy of further exploration.

A related implementation is one where 2D chiral samples are embedded in FP cavities in order to break the degeneracy between the clockwise and counter-clockwise rotating modes, without the need for strong coupling. Recent work has followed this approach by using a polystyrene layer under torsional shear stress, which under oblique incidence gives rise to 2D chiral effects emanating from an interference between LD and LB[35]. ACD samples draw from similar interferences, but instead exhibit LDLB interactions native to the sample itself, with strong 2D chiral selectivity at normal incidence, simplifying the experimental implementation of 2D chiral cavities, the principles of which we explored in a recent preprint[79]. We further wish to note recent theoretical studies predicting 3D chiral polaritons based on FP cavities with handedness-preserving mirrors, outlining a non-perturbative framework[80] as well as analytical solutions[81]. Altogether, these efforts pave the way to experimental realizations of chiral polaritonic phenomena, through which new technological opportunities are achievable.

Note added in proof: While this paper was in review, 2D chiral polaritons based on achiral FP cavities using ACD were experimentally reported for perovskite films[82].

## Methods

To determine the eigensolutions to Eq. (13), this Hamiltonian was expressed as a matrix in the photonic and molecular (electronic and vibrational) number bases. This matrix was then numerically diagonalized (Python 3.8, numpy version 1.22.3[83]), producing a set of eigenenergies and associated eigenvectors. These eigenvectors are then characterized by scalar metrics, as discussed in the main text.

Parameterization of the BDT-based oligothiophene is based on a combination of time-dependent density functional theory (TD-DFT) and spectral fitting as reported in previous work[47]. TD-DFT calculations were performed using QChem 5.3[84]. Two geometries (all-*cis* and all-*trans*) of the BDT-based oligothiophene molecule were first optimized in vacuum using the 6−311G(d,p) basis set and ωB97X-D functional[85]. Following this optimization, TD-DFT calculations were performed on the BDT-based oligothiophene molecule in vacuum and in chloroform using direct inversion of the iterative subspace[86] with CAM-B3LYP[87] as the functional and def2-TZVPPD[88] as the basis set of choice. The chloroform solvent was represented using an integral expansion formalism of a polarizable continuum model[89] with a dielectric constant of $\epsilon = 4.81$.

The oscillatory function $B_1(l)/l$ featured in the infinite-order Mueller calculus treatment was evaluated based on previous work by refs. 75,76. The underlying Mueller calculus formalism relies on Stokes vectors, representing the polarization properties of light. For a plane-wave electric field propagating in the $z$-direction as

$$\mathbf{E} = \begin{pmatrix} E_x e^{i\delta_x} \\ E_y e^{i\delta_y} \end{pmatrix}, \tag{25}$$

the Stokes vector is given by

$$\mathbf{S} = \begin{pmatrix} s_0 \\ s_1 \\ s_2 \\ s_3 \end{pmatrix} = \begin{pmatrix} E_x^2 + E_y^2 \\ E_x^2 - E_y^2 \\ 2E_x E_y \cos\delta \\ 2E_x E_y \sin\delta \end{pmatrix}, \tag{26}$$

with the phase difference $\delta \equiv \delta_y - \delta_x$. In order, the Stokes parameters appearing in this column vector correspond to total light intensity, $x - y$ intensity difference, $x' - y'$ intensity difference, where $x'$ is the bisector of the $x$ and $y$ axes and $y'$ is at a 90° clockwise rotation from $x'$, and circular polarization intensity difference (right-handed polarization minus left-handed polarization, or $r - l$). Within Mueller calculus, changes to light polarization due to interactions with a sample are governed by a linear transformation of the Stokes vector by a matrix, such that the final polarization is given by $\mathbf{S}_f = \mathbf{M}\mathbf{S}_0$, where $\mathbf{M}$ is the macroscopic "Mueller matrix" of the sample and $\mathbf{S}_0$ the initial polarization.

For a homogeneous sample, $\mathbf{M}$ relates trivially to its derivative $\mathbf{H} = \frac{d}{dz}\mathbf{M}$ by $\mathbf{M} = e^{-\mathbf{H}z}$, which decomposes as $\mathbf{H} = \bar{\alpha}\mathbf{I} + \mathbf{B} + \mathbf{D}$, where $\bar{\alpha}$ is the mean absorption and $\mathbf{B}$ and $\mathbf{D}$ relate the differential birefringence and dichroism, respectively[90]. Factoring out mean absorption, there are six unique polarization characteristics whose Lie algebras are

$$\mathbf{B} = \begin{pmatrix} 0 & 0 & 0 & 0 \\ 0 & 0 & -\beta_3 & \beta_2 \\ 0 & \beta_3 & 0 & -\beta_1 \\ 0 & -\beta_2 & \beta_1 & 0 \end{pmatrix}, \quad \mathbf{D} = \begin{pmatrix} 0 & d_1 & d_2 & d_3 \\ d_1 & 0 & 0 & 0 \\ d_2 & 0 & 0 & 0 \\ d_3 & 0 & 0 & 0 \end{pmatrix}. \tag{27}$$

Here, $\beta_1$, $\beta_2$, and $\beta_3$ represent linear ($x - y$), linear prime ($x' - y'$), and circular ($r - l$) birefringence, respectively, and $d_1$, $d_2$, $d_3$ are the analogs for dichroism.

Treating its anisotropic aspects according to their Lorentz group symmetries, the Mueller matrix to infinite order then follows as[75,76]

$$\mathbf{M} = \exp^{-\mathbf{H}l} = e^{-\alpha l}\mathbf{m} = e^{-\alpha l}\left(B_0\mathbf{I} + B_1(\mathbf{B} + \mathbf{D})^2 + B_2(-\mathbf{B} - \mathbf{D}) + B_3(\mathbf{D}_B - \mathbf{B}_D)\right). \tag{28}$$

Here, $l$ is the through-sample path length in the $z$-direction. Furthermore,

$$\mathbf{D}_B = \begin{pmatrix} 0 & \beta_1 & \beta_2 & \beta_3 \\ \beta_1 & 0 & 0 & 0 \\ \beta_2 & 0 & 0 & 0 \\ \beta_3 & 0 & 0 & 0 \end{pmatrix}, \quad \mathbf{B}_D = \begin{pmatrix} 0 & 0 & 0 & 0 \\ 0 & 0 & -d_3 & d_2 \\ 0 & d_3 & 0 & -d_1 \\ 0 & -d_2 & d_1 & 0 \end{pmatrix}, \tag{29}$$

are the incorporations of the dichroism and birefringence parameters onto the generators of birefringence and dichroism, respectively, and $B_i$ are $l$ dependent and real-valued "polarizance" parameters. These parameters detail the behavior of the restricted Lorentz group, SO$^+(1, 3)$, and follow upon combining $\mathbf{B}$ and $\mathbf{D}$, yielding

$$\begin{aligned} B_0 &= \left(\frac{R}{N}\right)^2 \cosh(Il) + \left(\frac{I}{N}\right)^2 \cos(Rl), \\ B_1 &= \left(\frac{1}{N}\right)^2 (\cosh(Il) - \cos(Rl)), \\ B_2 &= \left(\frac{1}{N}\right)^2 (R\sin(Rl) + I\sinh(Il)), \\ B_3 &= \left(\frac{1}{N}\right)^2 (I\sin(Rl) - R\sinh(Il)). \end{aligned} \tag{30}$$

Here, $N$, $R$, and $I$ are governed by

$$\sqrt{(\boldsymbol{\beta} + i\mathbf{d}) \cdot (\boldsymbol{\beta} + i\mathbf{d})} = N \exp\left[i\tan^{-1}\left(\frac{I}{R}\right)\right] = R + iI, \tag{31}$$

where $\mathbf{d} = (d_1, d_2, d_3)$ and $\boldsymbol{\beta} = (\beta_1, \beta_2, \beta_3)$ are the dichroism and birefringence vectors, respectively[76]. The polarizance parameter $B_1$ serves as the oscillating part of the function $B_1(l)/l$. Indeed, seeing that $N$, $I$, and $R$ are constants, the oscillatory behavior of $B_1$ becomes apparent. From the above, the approximations $B_0 \approx 1$ and $B_1 = \frac{1}{2}l^2$ taken for small path length $l$ are readily motivated (see Supplementary Methods 1), bringing Brown's treatment in line with the second-order Mueller calculus treatment.

As previously noted, within our quantum electrodynamical theory of ACD, the underlying Mueller calculus treatment considers cavity mirrors to represent reciprocal boundaries imposed on the sample. Importantly, within the applied boundaries, light of one handedness going forward corresponds to light of the opposite handedness going backwards. This is justified by noting that the (microscopic) Mueller matrix for forward (F) propagation $\mathbf{m}_F$ is equal to the matrix after a round trip incorporating two mirror reflections (M) and backwards (B) propagation, $\mathbf{m}_M\mathbf{m}_B\mathbf{m}_M$, given that only LD and LB contribute (see Supplementary Methods 2).

## Data availability

All relevant data is available at https://doi.org/10.5281/zenodo.10152500, which directs to github.com/andrewsalij/2DChiralPolACD[91]. Source data are provided with this paper.

## Code availability

The code used for simulations is available at https://doi.org/10.5281/zenodo.10152500, which directs to github.com/andrewsalij/2DChiralPolACD[91].

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

## Acknowledgements

This work was supported as part of the Center for Molecular Quantum Transduction (CMQT), an Energy Frontier Research Center funded by the U.S. Department of Energy, Office of Science, Basic Energy Sciences under Award No. DE-SC0021314.

## Author contributions

A.H.S. performed the calculations. R.T. proposed and oversaw the project. A.H.S., R.H.G., and R.T. contributed ideas and to the writing of the manuscript.

## Competing interests

The authors declare no competing interests.
