## [Peer Review File · Nature Communications]

Theory predicts 2D chiral polaritons based on achiral Fabry–Pérot cavities using apparent circular dichroismREVIEWER COMMENTS

Reviewer #1 (Remarks to the Author):

The paper by Salij et al. discusses some very interesting ideas regarding the realization of chiral polariton states. However, the paper

has several shortcomings in the way the ideas are presented. I would suggest the authors rewrite the paper and clearly present the

main ideas about storing chirality in the cavity clearly in the beginning. The paper is not easy to read and understand, even for me that

has worked in the field for a long time.

I have some concrete questions about the work.

1. The authors claim their approach is non-perturbative - this statement is confusing - as the model they use is basically Jaynes-Cummings model

and increasing the size of this matrix does not make the approach non-perturbative - in my view.

2. The Hamiltonian in Eq. (2) only includes dipole transition moments - thus not able to discriminate enantiomers - would the effects discussed

not disappear when orientationally averaged?

3. Are the ideas discussed in Ref.35 by Gautier et al not similar - a detailed analysis should be presented.

4. Recent advances in the full treatment of polaritons in chiral cavity fields have not been cited. See for instance

<https://doi.org/10.48550/arXiv.2209.01987>

Unless the authors significantly improve the presentation, this paper would be more appropriate in a lower-impact journal.

Reviewer #2 (Remarks to the Author):

The manuscript presents a theoretical study of the possibility of using a medium which presents ACD, one of the observables of 2D chirality, to generate chiral polaritons. To this end the paper first introduces the ACD as an observable within the rotating wave approximation and give a detailed description of the different approximation and their implication. This description is then applied to chiral polaritons, and a three-state approximation is used to understand complex features of the dispersion curves. Using the previously detailed analysis a set of rules is drawn to generate chiral polaritons and the case of using a type of oligomer is detailed.

Overall, the manuscript is good and raises an interesting question that it answers gracefully. This work lays very quite general rules and could have an excellent reach for various fields. I particularly think of the work of polaritonic chemistry which has seen an increasing amount of paper published recently but also in the field of chiral sensing. Nonetheless I had a few questions and comments regarding the main manuscript.

1. If I correctly understand the discussion of the second section lies outside the realm of strong coupling and the only use of the cavity mode is to bound the length of the cavity and so the result are described in an off-resonance state. Would the intrinsic polarization of the cavity mode (i.e. TE, TM) impact the off-resonance result? It seems that the cavity mode is only used to define a resonance frequency and a length to the cavity, but wouldn't it also interact weakly with the system?
2. Regarding the ACD, is there also a requirement regarding how close the two transition needs to be in energy to see the rise of ACD?
3. The TSA approximation is a little bit troubling for me. There is the following statement p. 15: "Within these spectral regions $g_{\alpha}/\Gamma(\alpha) = \sigma$ offers a direct manner by which one can tune the polaritonic chirality." If you want to integrate chirality as a photonic dof through strong coupling, you then need to have minimum photonic mixing so that you can tune polaritonic chirality. This seems to be quite counterintuitive.
4. Regarding the lowest chiral polaritons ($\alpha=1$), we see that in Fig.3a and Fig.3c there is another change of the sign of g_{α} at 3 eV. From eq 22 the change that we are seeing reflect the change of sign of the chiral interaction term which from the reasoning p. 15 is because of the photonic mixing.
5. I am not sure to understand why $g_{\alpha=1}$ is so different than $g_{\alpha=2}$ far away from the photonic line. Shouldn't we recover the same behavior but inverse for $\alpha=1/2$?

6. If I am not mistaken, could we obtain an opposite g_{α} by setting $\text{Beta}_{21} = -45^\circ$? If so it would maybe be interesting to compare the dispersion curve in the SI.

7. The last comment is maybe outside of the scope of this manuscript as you deal with the lowest energy cavity mode but how does the intracavity mode profile affect the g_{α} coefficient?

Reviewer #3 (Remarks to the Author):

This manuscript presents a study of what the authors refer to as “chiral polaritons” formed by coupling circularly-polarized optical modes of an ordinary Fabry-Perot cavity to an ensemble of resonant non-chiral emitters featuring so called “apparent circular dichroism”. The authors develop a Hamiltonian model of this interaction and analyze the energy spectra of the resulting polaritonic states. While this is an intriguing work, and the study of chiral states of electromagnetic field is a hot area of nanophotonics gaining a lot of steam, the manuscript lacks clarity and is rather difficult to follow.

1. As long as none of the constituents of the system (neither optical cavity nor molecular emitters) are chiral to begin with, why do the authors go on to call the resulting polaritonic state chiral? This is a highly misleading move.
2. Could the authors be more specific about what these quantum emitters with ACD are exactly? Can these be atoms, molecules, complicated harmonic meta-atoms? What are the geometry requirements for a primitive emitter to exhibit ACD? The authors do provide some molecular examples in section 6, but I think the manuscript would benefit a lot if the authors describe right in section 1 what these emitters are from the microscopic point of view (second paragraph on p. 3 seems like a good place for that).
3. The same paragraph on p. 3 describes ACD as “a differential absorption of left-handed and right-handed circularly-polarized light”. Again, this is rather misleading definition, because this ACD phenomenon should be entirely insensitive to the handedness, since the sample is not chiral. Instead, ACD should be sensitive to the direction of polarization rotation of circularly polarized light, which can be expressed by the spin angular momentum density of incident electromagnetic field.
4. The caption of Fig 1 mentions purple arrow, while there are no purple objects in the figure.
5. On p. 5 the authors introduce two transition dipole moments (μ_+ and μ_-) of a given electronic transition of a given emitter for two orthogonal polarizations. There can only be a single transition dipole moment for a particular transition of a quantum system: it is an internal quantity of the emitter that is completely independent of the surrounding electromagnetic field. Can the authors be more clear about what they mean by distinguishing these two quantities?

6. On p. 6 the authors bring up so called “chiral interaction term” in Eq. 8. I realize this is pure semantics, but as long as the underlying system has nothing to do with chirality, I believe it is misleading to use a term like that.

7. From Eq. 12 the theory essentially looks like the standard Jaynes-Cummings Hamiltonian but implemented for two circularly polarized cavity modes, and a quantum emitter with a complicated ladder of dipole transitions oriented at different angle with respect to each other. It appears that it is this multi-transition structure of the emitter, which is the key origin of the ACD phenomenon, and perhaps it should be visualized graphically in the very first figure of the manuscript.

8. The spectra presented in Fig. 6 are the results of the quantum-mechanical Hamiltonian of the multi-resonant ACD system coupled to circularly-polarized modes of an optical cavity. What is the significance of the high-frequency dielectric constant ϵ_{∞} in all this?

After all, this is a nice theoretical manuscript presenting a potentially useful analytical model. Nonetheless, even with the above comments addressed, the manuscript would still not be suitable for publication in Nat. Commun., as the results presented in it appear to me rather incremental in nature, and do not describe a qualitatively novel optical phenomenon. And this manuscript definitely does not describe what it claims to – namely, the phenomenon of chiral polaritons.

Responses to Reviewer #1

The paper by Salij et al. discusses some very interesting ideas regarding the realization of chiral polariton states. However, the paper has several shortcomings in the way the ideas are presented. I would suggest the authors rewrite the paper and clearly present the main ideas about storing chirality in the cavity clearly in the beginning. The paper is not easy to read and understand, even for me that has worked in the field for a long time.

We thank the reviewer for taking the time to carefully examine our manuscript, and for underscoring the interest of the ideas explored in our submitted work. We furthermore appreciate the reviewer's comments regarding the lack of clarity in presenting those ideas. We have made significant edits throughout our manuscript in order to improve the presentation. In particular, the revised Introduction begins to present the idea of addressing and storing chirality in the cavity, which we believe greatly helps guiding the subsequent introductory narrative.

I have some concrete questions about the work.

1. The authors claim their approach is non-perturbative - this statement is confusing - as the model they use is basically Jaynes-Cummings model and increasing the size of this matrix does not make the approach non-perturbative - in my view.

The Reviewer raises the point that the Jaynes–Cummings model is not truly a nonperturbative approach, which we agree with. We appreciate how our phrasing in the Introduction, where we contrast our method with the "perturbative" approach from our previous work, may be confusing in this regard. To remedy this, we now refer to our previous work as "semiclassical" in order to avoid this confusion.

In the revised Introduction we have modified paragraph 4 as follows:

In a recent work, we presented a microscopic yet ~~perturbative~~ semiclassical theory of ACD under weak light-matter coupling wherein the Mueller calculus treatment embodied by Eq. 1 was combined with a

We have furthermore modified paragraph 5 as follows:

In demonstrating 2D chiral polaritons, the present Paper introduces a quantum electrodynamical theory of ACD. Departing from the ~~perturbative-semiclassical~~ treatment from our previous work,⁴⁸ which ~~prohibited-describing strong coupling fails to describe strong coupling and polariton formation~~, the current theory is based on an appropriately-extended Jaynes–Cummings model,⁷⁴ ~~considering a~~ providing a quantum-mechanical treatment of a single quantum emitter in an ideal (lossless) cavity. It is shown that

2. The Hamiltonian in Eq. (2) only includes dipole transition moments - thus not able to discriminate enantiomers - would the effects discussed not disappear when orientationally averaged?

The reviewer correctly points out that an orientational average over all three dimensions will eliminate the ACD effect. The effect is therefore only present for oriented samples, where it should be noted that a rotational average in the sample plane (normal to the light propagation direction) does not diminish the effect, as discussed in our previous work [Ref. 48]. We have clarified this in the revised manuscript, in the second paragraph of the subsection entitled "2D chiral interaction terms":

-, and polarization (which therefore obeys the same inversion antisymmetry). Notably, achiral as well as 3D chiral samples have $\tilde{\mu}_{n,+} = \tilde{\mu}_{n,-}$, as a result of which there is no chiral-selectivity with regard to the circularly-polarized $\lambda = +$ and $\lambda = -$ polarization modes of the FP cavity. This selectivity will similarly vanish for a full three-dimensional orientational average of an ACD sample. Indeed, the ACD phenomenon relies on oriented samples, although rotations in the plane normal to the light propagation direction do not diminish its effect ⁴⁸.

3. Are the ideas discussed in Ref.35 by Gautier et al not similar - a detailed analysis should be presented.

In Ref. 35, Gautier et al. propose a related implementation where a FP cavity is made chiral by embedding a polystyrene layer under torsional shear stress, which under oblique incidence gives rise to 2D chiral effects by drawing from a similar interference between LD and LB as the LDLB interactions underlying ACD. We have added a discussion on the similarities and differences between these two phenomena in the fourth paragraph of the Discussion section, and included a mentioning of a preprint where we harnessed ACD for chiral FP cavity engineering in similar spirit to that study, as follows:

A related implementation is one where 2D chiral samples are embedded in FP cavities in order to break the degeneracy between the clockwise and counter-clockwise rotating modes, without the need for strong coupling. Recent work has followed this approach by using a polystyrene layer under torsional shear stress, which under oblique incidence gives rise to 2D chiral effects emanating from an interference between LD and LB ³⁵. ACD samples draw from similar interferences, but instead exhibit LDLB interactions native to the sample itself, with strong 2D chiral selectivity at normal incidence, simplifying the experimental implementation of 2D chiral cavities, the principles of which we explored in a recent preprint ⁸³. We

4. Recent advances in the full treatment of polaritons in chiral cavity fields have not been cited. See for instance <https://doi.org/10.48550/arXiv.2209.01987>

We thank the reviewer bringing this reference to our attention. We have added to the Discussion section a mentioning of this work (included as Ref. 84), as follows:

implementation of 2D chiral cavities, the principles of which we explored in a recent preprint⁸³. We further wish to note recent theoretical studies predicting 3D chiral polaritons based on 3D-chirality-by-the-use-of-FP-cavities-with handedness-preserving mirrors, outlining a nonperturbative framework⁸⁴ as well as analytical solutions.⁸⁵ Altogether, these efforts pave the way to experimental realizations of chiral polaritonic phenomena, through which new technological opportunities are achievable.

Unless the authors significantly improve the presentation, this paper would be more appropriate in a lower-impact journal.

We thank the reviewer again for the thoughtful examination of our manuscript, and hope that he/she our revisions to have appropriately improved the presentation of our findings.

Responses to Reviewer #2

The manuscript presents a theoretical study of the possibility of using a medium which presents ACD, one of the observables of 2D chirality, to generate chiral polaritons. To this end the paper first introduces the ACD as an observable within the rotating wave approximation and give a detailed description of the different approximation and their implication. This description is then applied to chiral polaritons, and a three-state approximation is used to understand complex features of the dispersion curves. Using the previously detailed analysis a set of rules is drawn to generate chiral polaritons and the case of using a type of oligomer is detailed.

Overall, the manuscript is good and raises an interesting question that it answers gracefully. This work lays very quite general rules and could have an excellent reach for various fields. I particularly think of the work of polaritonic chemistry which has seen an increasing amount of paper published recently but also in the field of chiral sensing. Nonetheless I had a few questions and comments regarding the main manuscript.

We thank the reviewer for the careful examination of our manuscript, and for underscoring the importance of our work. We furthermore thank the reviewer for raising a series of insightful questions and comments, which have enabled us to further improve the presentation of our results.

1. If I correctly understand the discussion of the second section lies outside the realm of strong coupling and the only use of the cavity mode is to bound the length of the cavity and so the result are described in an off-resonance state. Would the intrinsic polarization of the cavity mode (i.e. TE, TM) impact the off-resonance result? It seems that the cavity

mode is only used to define a resonance frequency and a length to the cavity, but wouldn't it also interact weakly with the system?

The reviewer correctly points out that apparent circular dichroism (ACD) is treated within the weak light-matter coupling regime in parts of the subsection entitled "2D chiral interaction terms" (previously the second section). The reviewer's questions have led us to realize that a few further clarifications should be added to these parts. First, this treatment is agnostic to the nature of the optical modes, which could represent bound modes inside a cavity as well as free-field modes (as applicable to absorption spectroscopy). What this treatment allowed us to do is to derive effective transition dipole moments for modes with (2D) chiral polarization at a given frequency, through incorporation of the effect of the birefringent light-matter interaction. These dipole moments are then incorporated in a Jaynes–Cummings-like Hamiltonian involving a discrete mode (or, rather, two degenerate and orthogonally-polarized discrete modes). This Hamiltonian instead does assume the presence of a cavity with a set length bound by the mode as well as a well-defined resonance frequency. We should further clarify that we assume the light propagation direction to be exactly normal to the plane of an ideal cavity at all times. With the two cavity modes being fully degenerate and orthogonally polarized, any arbitrary superposition of these modes yields an intrinsic polarization of the cavity. Such symmetry is broken once an ACD material is embedded, as a result of which polariton modes emerge that have a well-defined polarization, as quantified in the subsection entitled "2D chiral polaritons".

In the revised manuscript we have modified the second paragraph of the subsection entitled "2D chiral interaction terms" as follows:

event. Our approach to incorporate ACD in this Hamiltonian is to only explicitly invoke the (second) dichroic interaction, and to include the (first) birefringent interaction through its effect on the transition dipole ~~moment projection~~ $\tilde{\mu}_{n,\lambda}$. In doing so, we first proceed to ~~consider ACD within the perturbative treat~~ ACD semiclassically and within the regime of weak light-matter interaction regime. ~~Within this regime, we have previously shown, for coupling. This treatment is agnostic to the presence or absence of a cavity, and merely considers ACD as a chiroptical response resulting from light with a given optical frequency incident on a sample of a given thickness. Within this treatment, and assuming~~ molecular crystals where intermolecular interactions are negligible, we have previously shown that the ACD transition rate takes the form⁴⁸

We furthermore modified the first paragraph of the subsection entitled "2D chiral polaritons", as follows:

We will proceed to evaluate the polaritonic states arising from ACD samples embedded in ~~achiral-FP cavities. In doing so, an idealized achiral FP cavity. Assuming the light propagation direction to be normal~~

to the cavity plane at all times, the relevant optical modes organize in degenerate pairs with orthogonal polarization. Such cavity by itself does not break the symmetry between the two modes, and as such modes within any orthogonal polarization basis will form valid intrinsic modes of the cavity. Embedding of an ACD sample breaks this symmetry within the basis of $\lambda = +$ and $\lambda = -$ polarized modes. To describe this, we extend the light-matter interaction Hamiltonian from Eq. 2 within the single-molecule limit in order to

2. Regarding the ACD, is there also a requirement regarding how close the two transition needs to be in energy to see the rise of ACD?

In our current manuscript, quantities related to ACD are normalized to total absorption, as a result of which the rise of ACD itself is a bit obscured. However, as we have shown in a previous paper (Ref. 48), the two transitions need to be within each other's linewidth in order for ACD itself to be appreciable. We have added a mentioning of this to the seventh paragraph of the subsection entitled "2D chiral interaction terms".

pearing in Eq. 11 are identical. We note that whereas the 2D chiral interaction terms are proportional to ACD normalized to total absorbance, we previously analyzed the dependence of ACD itself on the frequency gap, and found that for ACD to be appreciable the involved transitions should be within each other's linewidth

48

3. The TSA approximation is a little bit troubling for me. There is the following statement p. 15: "Within these spectral regions $g\omega/l(n) = \sigma n$ offers a direct manner by which one can tune the polaritonic chirality." If you want to integrate chirality as a photonic dof through strong coupling, you then need to have minimum photonic mixing so that you can tune polaritonic chirality. This seems to be quite counterintuitive.

We thank the reviewer for raising this point. Upon closer inspection, we concluded that the entire paragraph lacked clarity, and unnecessarily introduced contradictory arguments, as the reviewer rightfully pointed out. The simple version of this paragraph is that the TSA is seen to hold up well compared to the full Hilbert space results, except when close to the light line, where a predominant photonic component mixes equally to both excited states (both being off resonant). As such, the TSA provides a useful (albeit approximate) guideline with which to tune 2D chirality away from the light line.

We have fully rewritten the fourth paragraph of the subsection entitled "Three-state approximation" in order to better reflect this, while also correcting for a (factor of 2) typo in the relationship between g and σ , as follows:

In order to assess the accuracy of the TSA, we compare in Fig. 5 ~~results~~ TSA results for $n = 1$ against the numerical solutions of the full Hilbert space for the $A_0 = 35 \text{ eV}/ec$ case previously considered in Fig. 3. ~~Shown are the polariton dispersions and \bar{g}^α as a function of Ω . The TSA is taken for $n = 1$. The resulting~~ are the polariton dispersions and the polaritonic dissymmetry factors. The dispersions are indistinguishable from those predicted by the full Hilbert space, indicating that mixing due to $n = 2$ has a negligible impact on $E_{(n=1)}^{u/l}$. In contrast, discrepancies are observed for the polaritonic dissymmetry factors. Within the TSA

the upper and lower polariton pair have identical 2D chirality, $g_{(n=1)}^u = g_{(n=1)}^l$, while the overall polaritonic dissymmetry is seen to be monosignate. Within the full Hilbert space, $g_{(n=1)}^u = g_{(n=1)}^l$ is violated ~~, with~~ discrepancies occurring exclusively for polaritonic states close to the light line (for which $E_{(n=1)}^{u/l} \approx \Omega$), giving rise to ~~the bisignate profile of \bar{g}^α obtained within the bisignate dissymmetry factors. Discrepancies~~ between the TSA and the full Hilbert space ~~. This behavior is~~ results are rationalized by the two excited states of the sample coupling only indirectly to one another through the photonic states. When ~~close to~~ separated ~~from~~ the light line polaritonic states contain a predominant photonic component that couples directly to both excited states, ~~as a result of which $g_{(n)}^{u/l} \neq \sigma_n$, whereas when separated from the light-line polaritons consist~~ predominantly of a single excited state with only a minor photonic component, and thus weak mixing of, polaritonic states have a substantial matter contribution due to a single resonant excited state, and the other excited state ~~, as a result of which the TSA is accurate. Within these spectral regions $g_{(n)}^{u/l} = \sigma_n$ offers a~~ direct manner by ~~mixes only weakly through the photonic component. As a result, $g_{(n)}^{u/l} = 2\sigma_n$ governs the~~ polaritonic dissymmetry, rendering this relationship a useful (albeit approximate) manner with which one can tune ~~the polaritonic chirality~~ 2D chirality. Close to the light line, however, polaritonic states instead contain a predominant photonic component that couples more equally to both excited states (both being offresonant), ~~as a result of which $g_{(n)}^{u/l} \neq 2\sigma_n$, rendering the TSA inaccurate.~~

4. Regarding the lowest chiral polaritons ($\alpha=1$), we see that in Fig.3a and Fig.3c there is another change of the sign of g_α at 3 eV. From eq 22 the change that we are seeing reflect the change of sign of the chiral interaction term which from the reasoning p. 15 is because of the photonic mixing.

The reviewer rightfully points out that the sign change of the (polaritonic) dissymmetry factor at 3 eV is due to the change of sign of the 2D chiral interaction term. We have reworded the relevant text in order to better convey this. Accordingly, the last paragraph of the subsection entitled "Three-state approximation" has been changed as follows:

$\chi^{\alpha=1}$ monotonically decreases away from the resonance $\Omega \approx \omega_1$. ~~Outside the~~ We also note that the sign change of $\bar{g}^{\alpha=1}$ at $\sim 3 \text{ eV}$ is due to a change of sign of σ_1 . However, at $\sim 2 \text{ eV}$, which lies outside the TSA regime, the sign change of $\bar{g}^{\alpha=1}$ ~~(giving rise to the bisignate profile) is a direct consequence of a~~ is instead due to a mixing of the $n = 2$ excited state. The TSA analysis furthermore confirms that the A_0

5. I am not sure to understand why $g_{\alpha=1}$ is so different than $g_{\alpha=2}$ far away from the photonic line. Shouldn't we recover the same behavior but inverse for $\alpha=1/2$?

Here, the reviewer may be referring to the inverted behavior observed for the 2D chiral interaction terms in Fig. 2 (a), meaning $\sigma_1(\Omega - \omega_1) \approx \sigma_2(\omega_2 - \Omega)$, which according to the TSA should be reflected in the polaritonic dissymmetry factors for $\alpha=1$ and $\alpha=2$ in their respective regions away from the light line. In Fig. 3 this is indeed borne out. We have added a clarification to the last paragraph of the subsection entitled "Three-state approximation", as follows:

$\tilde{g}^{\alpha=1} \approx 2\sigma_1$, which is readily verified through a comparison with Fig. 2 (b-d). We further note that in Fig. 3, $\sigma_1(\Omega - \omega_1) \approx \sigma_2(\omega_2 - \Omega)$ governs the behaviors of $\tilde{g}^{\alpha=1}$ and $\tilde{g}^{\alpha=2}$ in their respective regions away from the light line (which follows from an application of the TSA to both excited states). Lastly, it should be pointed

6. If I am not mistaken, could we obtain an opposite g_{α} by setting $\beta_{21}=-45^\circ$? If so it would maybe be interesting to compare the dispersion curve in the SI.

We thank the reviewer for making the suggestion to include comparative dispersion curves for $\beta_{21} = -45^\circ$. We have added Section S6 to the SI where such comparison is presented. As the reviewer suspected, the dissymmetry factors come out inverted. This can be understood by the principle that such angle inversion effectively representing a sample inversion, which for 2D chiral samples returns their enantiomer. We have added a mentioning of this in the last paragraph of the subsection entitled "Three-state approximation", as follows:

light line (which follows from an application of the TSA to both excited states). Lastly, it should be pointed out that the polaritonic dissymmetry undergoes a global sign change upon inverting the inter-dipole angle

(see supplementary information). Such is evident from the analytical expressions within the TSA (seeing that it depends on the 2D chiral interaction terms, and cf. Eq. 11), and can generally be understood by the principle that such angle inversion represents a sample flipping, which for 2D chiral samples returns their (2D) enantiomer.

7. The last comment is maybe outside of the scope of this manuscript as you deal with the lowest energy cavity mode but how does the intracavity mode profile affect the g_{α} coefficient?

The reviewer's question hints at the possibility of impacting the polaritonic dissymmetry factor through shaping the mode profile. Although this is indeed beyond the scope of the current manuscript, which only considers the lowest mode, we have included a mentioning of this possibility in the second paragraph of the Discussion section.

suppressed in the synthesis of ACD samples. It will also be of interest to consider resonances between ACD samples and higher-energy modes of a cavity, and to assess whether polaritonic dissymmetry factors can be enhanced or suppressed through mode profiles.

Responses to Reviewer #3

This manuscript presents a study of what the authors refer to as “chiral polaritons” formed by coupling circularly-polarized optical modes of an ordinary Fabry-Perot cavity to an ensemble of resonant non-chiral emitters featuring so called “apparent circular dichroism”. The authors develop a Hamiltonian model of this interaction and analyze the energy spectra of the resulting polaritonic states. While this is an intriguing work, and the study of chiral states of electromagnetic field is a hot area of nanophotonics gaining a lot of steam, the manuscript lacks clarity and is rather difficult to follow.

We thank the reviewer for taking the time and effort to carefully examine our manuscript, and for underscoring the importance of studying chiral optical states. We furthermore thank the reviewer for contributing insightful comments, and for pointing out ways to improve the clarity of our manuscript.

1. As long as none of the constituents of the system (neither optical cavity nor molecular emitters) are chiral to begin with, why do the authors go on to call the resulting polaritonic state chiral? This is a highly misleading move.

We greatly appreciate this comment, as it made us realize our manuscript lacked specificity as to what we meant by "chiral". Chiral oftentimes refers to "3D chirality", which is associated with point reflections, and which manifests in a three-dimensional structure (such as a molecule). It is true that none of the constituents of the system exhibits 3D chirality. Instead, the polaritonic states exhibit "2D chirality", which instead is associated with plane reflections. For the photonic part of the polaritons, this 2D chirality behaves like angular momentum states polarized in the two-dimensional plane normal to the light-propagation direction. For the ACD samples, 2D chirality arises due to their oblique and nondegenerate transitions which are oriented with respect to the light-propagation direction.

It was certainly not our intention to use chirality in a misleading way. Rather, we intended to conform our nomenclature to the literature that previously described ACD as chiral. However, we appreciate that this could be highly confusing to the reader. In order to remedy this, in our revisions we have consistently specified "2D chirality" throughout the manuscript, and including the title. We hope that the reviewer finds these changes to be appropriate.

2. Could the authors be more specific about what these quantum emitters with ACD are exactly? Can these be atoms, molecules, complicated harmonic meta-atoms? What are the geometry requirements for a primitive emitter to exhibit ACD? The authors do

provide some molecular examples in section 6, but I think the manuscript would benefit a lot if the authors describe right in section 1 what these emitters are from the microscopic point of view (second paragraph on p. 3 seems like a good place for that).

We agree that more specificity on the quantum emitters exhibiting ACD will be helpful. As it turns out, ACD arises for samples consisting of oriented molecules featuring a pair of bright, non-degenerate, and oblique transition dipoles. We have detailed this in the fourth paragraph of the Introduction, as reproduced below. We have further included in Fig. 1 a schematic of this geometric requirement, also following up on point 7 by the reviewer, which will further help clarifying the quantum emitters of interest.

Lorentz oscillator model⁴⁸. This allowed ~~identification of the (supra)molecular design rules for optimizing ACD within the linear regime~~ to identify geometric sample properties that enable this process, namely the presence of oriented molecules featuring a pair of bright, nondegenerate, and oblique transition dipoles, as depicted in Fig. 1 (a,b). This work was motivated by a recent series of spectroscopic studies on ACD in

3. The same paragraph on p. 3 describes ACD as “a differential absorption of left-handed and right-handed circularly-polarized light”. Again, this is rather misleading definition, because this ACD phenomenon should be entirely insensitive to the handedness, since the sample is not chiral. Instead, ACD should be sensitive to the direction of polarization rotation of circularly polarized light, which can be expressed by the spin angular momentum density of incident electromagnetic field.

We thank the reviewer for raising this comment. In the revised manuscript, we have removed references to "circular polarization" where appropriate, as we agree that the modes inside the FP cavity should be thought of as clockwise and counter-clockwise rotating optical fields. We left a mentioning of circularly-polarized absorption for a given irradiation direction, since that corresponds to how ACD has originally manifested in the chiroptics literature. We have revised the relevant paragraph as follows:

in interest.^{43,44} It ~~is a results from oblique dichroic and birefringent axes of a sample, giving rise to a net~~

~~difference in the absorption of clockwise and counter-clockwise rotating optical fields [corresponding to a differential absorption of left-handed and right-handed circularly-polarized light resulting from a macroscopic linear orientation of non-parallel dichroic and birefringent axes of a sample, for a given irradiation direction^{42,45}~~ as depicted in Fig. 1 (a)^{42,45}(c)]. Within Mueller calculus, where light-matter interactions are described by

4. The caption of Fig 1 mentions purple arrow, while there are no purple objects in the figure.

We thank the reviewer for spotting this mistake. We have changed "purple" to "green" in the caption.

5. On p. 5 the authors introduce two transition dipole moments ($\mu_{n,+}$ and $\mu_{n,-}$) of a given electronic transition of a given emitter for two orthogonal polarizations. There can only be a single transition dipole moment for a particular transition of a quantum system: it is an internal quantity of the emitter that is completely independent of the surrounding electromagnetic field. Can the authors be more clear about what they mean by distinguishing these two quantities?

We apologize for the confusing explanation in our previous submission. The reviewer rightfully points out there can only be a single transition dipole moment for a given transition n . Instead, $\mu_{n,+}$ and $\mu_{n,-}$ are simply the projections of the transition dipole vector onto the + and - polarizations. In the revised manuscript we have changed the wording accordingly, and the relevant sentence has been modified as follows:

In Eq. 2, n runs over the excited states of the sample, with \hat{b}_n^\dagger and \hat{b}_n as the corresponding creation and annihilation operators, respectively, ~~and with~~. Moreover, $\hbar\omega_n$ ~~is~~ the associated excited state energy (where the ground state energy is taken to be zero as a reference). ~~The~~, and $\tilde{\mu}_{n,\lambda}$ ~~is the projection of the associated transition dipole moment $\tilde{\mu}_{n,\lambda}$ is defined specifically for vector onto~~ the λ circular polarization

~~and~~ polarization (which therefore obeys the same inversion antisymmetry). Notably, achiral as well as

6. On p. 6 the authors bring up so called "chiral interaction term" in Eq. 8. I realize this is pure semantics, but as long as the underlying system has nothing to do with chirality, I believe it is misleading to use a term like that.

We agree with the reviewer, for as much as this term has nothing to do with 3D chirality. Consistent with our response to point 1, we have changed the nomenclature to "2D chiral interaction term" throughout the revised manuscript.

7. From Eq. 12 the theory essentially looks like the standard Jaynes-Cummings Hamiltonian but implemented for two circularly polarized cavity modes, and a quantum emitter with a complicated ladder of dipole transitions oriented at different angle with respect to each other. It appears that it is this multi-transition structure of the emitter, which is the key origin of the ACD phenomenon, and perhaps it should be visualized graphically in the very first figure of the manuscript.

It is indeed the multi-transition structure of the emitter that is the key origin of the ACD phenomenon, the simplest example of which consists of two bright, nondegenerate, and oblique transition dipoles. We have included an illustration of this example in Fig. 1, and updated the caption, as reproduced below.

8. The spectra presented in Fig. 6 are the results of the quantum-mechanical Hamiltonian of the multi-resonant ACD system coupled to circularly-polarized modes of an optical cavity. What is the significance of the high-frequency dielectric constant ϵ_{∞} in all this?

The high-frequency dielectric constant, which relates to the refractive index of the sample, impacts the effective pathlength. As it turns out, it has no impact within the second-order Mueller calculus treatment (see Eq. 10), but it (weakly) impacts the infinite-order treatment such as presented in Fig. 6. We have added a clarification to the last paragraph of the subsection entitled "2D chiral interaction terms".

ideal FP cavity confinement. For the remainder of this Paper, we exclusively employ ϵ_{∞} and ν have no impact on the first-order treatment, but that they do have a (minor) effect on the infinite-order 2D chiral interaction terms, which we employ for the remainder of this Paper.

After all, this is a nice theoretical manuscript presenting a potentially useful analytical model. Nonetheless, even with the above comments addressed, the manuscript would still not be suitable for publication in Nat. Commun., as the results presented in it appear to me rather incremental in nature, and do not describe a qualitatively novel optical

phenomenon. And this manuscript definitely does not describe what it claims to – namely, the phenomenon of chiral polaritons.

We thank the reviewer again for thoroughly inspecting our manuscript, and for contributing insightful comments. By having followed up on these comments, we believe to have substantially improved the presentation of our study, and to have clarified the phenomenon in question which is more appropriately referred to as "2D chiral polaritons". However, the reviewer also suggests that our work is to some degree incremental, and we would like to directly refute this issue. To the best of our knowledge, our work is the first to present the theoretical underpinnings of 2D chiral polaritons, and to discuss a practical route towards their realization based on apparent circular dichroism. In the last paragraph of the Discussion section we added additional discussion of their potential applications, which warrant upcoming experimental and engineering efforts aimed at realizing 2D chiral polaritons. We should note that a similar interest is invested in "3D" chiral polaritons whose challenges and opportunities are partly orthogonal to 2D chiral polaritons, but which are arguably more difficult to experimentally realize (due to the necessity of handedness-preserving mirrors or ring cavities). As such, we expect upcoming developments to be particularly expeditious for 2D chiral polaritons. Our submission lays the groundwork for this line of inquiry, and as such we believe that it bears a high level of novelty.

Other changes made to the manuscript

- Changes have been made in order to comply with the Nature Communications guidelines.
- Various typos have been corrected.

REVIEWERS' COMMENTS

Reviewer #2 (Remarks to the Author):

Following the first round of the review the manuscript did an overall of part of the paper, such as the three-state approximation, and rewrote and added sections where needed. This simplified and de-convoluted some parts which were, at first, difficult to understand.

The response to my questions were answered, and I acknowledged the other reviewer's excellent question and remarks. The essence of the paper which is to utilize 2D chirality as a tool to implement chirality in polaritonic systems and the design of general rules is, to my knowledge, never formally discussed this way.

Reviewer #3 (Remarks to the Author):

In this revision of their manuscript "2D chiral polaritons based on achiral Fabry–Perot cavities using apparent circular dichroism" Salij et al., in my opinion, have addressed technical and specific points raised by all three referees, and improved presentation of the manuscript to some degree. The conclusions are original, and overall this a well-executed theoretical study.

Nevertheless, I have to point out that, in my opinion, this manuscript does not offer the degree of novelty and significance that would justify its publication in Nature Communications. Although it is the first work to present the theoretical underpinnings of "2D chiral polaritons", as the authors claim, I find the system they analyze rather synthetic. The findings about this class of optical systems, in my opinion, do not present the kind of scientific knowledge and findings that typically merits publication in Nature Communications.

The manuscript does create an impressions of a very well written research article, which would be suitable for a more specialized journal.